# Mutations in the kinesin KIF12 promote MASH in humans and mice by disrupting lipogenic enzyme turnover

Asieh Etemad [ID][1], Yosuke Tanaka [ID][1], Shuo Wang [ID][1], Mordechai Slae[2], Mutaz Sultan[3], Orly Elpeleg[2] & Nobutaka Hirokawa [ID][1,4 ✉]

## Abstract

As a common cause of liver cirrhosis, metabolic dysfunction-associated steatohepatitis (MASH) is regarded as a target of therapeutic intervention. However, a successful therapy has not yet been found, partly because the molecular pathogenesis is largely elusive. Here we show that KIF12 kinesin suppresses MASH development by accelerating the breakdown of two lipid bio-synthesis enzymes, acetyl-CoA carboxylase 1 (ACC1) and pyruvate carboxylase (PC), in hepatocytes. We report three familial early-onset liver cirrhosis pedigrees with homozygous KIF12 mutations, accompanying MASH-like steatosis and cholestasis. The mouse genetic model carrying the corresponding *Kif12* nonsense mutation faithfully reproduced the phenotypes as early as between 8 and 10 weeks of age. Furthermore, KIF12-deficient HepG2 cells exhibited significant steatosis, which was ameliorated by overexpressing a proline-rich domain (PRD) of KIF12. We found that KIF12-PRD promotes the degradation of ACC1 and PC, and this effect is likely to be through its direct interaction with these enzymes. Interestingly, KIF12 enhanced the ubiquitination of ACC1 by the E3 ligase COP1 and colocalized with these proteins as seen by super-resolution microscopy imaging. These data propose a role for KIF12 in suppressing MASH by accelerating turnover of lipogenic enzymes.

**Keywords** KIF12; MASH; ACC1; Cirrhosis; Mouse Model
**Subject Categories** Cell Adhesion, Polarity & Cytoskeleton; Metabolism; Molecular Biology of Disease

## Introduction

In developed countries, metabolic dysfunction-associated fatty liver disease (MAFLD) and metabolic dysfunction-associated steatohepatitis (MASH) constitute a very common disease entity of ca. 30% prevalence, becoming an emerging major threat of health (Huang et al, 2021). Several percentages in MASH patients should develop cirrhosis and hepatocellular carcinoma (HCC), which is the third leading cause of death in people of 45–64 years old (Gines et al, 2021). MAFLD raises the comorbidity of type 2 diabetes (Targher et al, 2021), suggesting a bidirectional relationship with metabolic syndrome (Jensen et al, 2018; Lau et al, 2017; Mells et al, 2015; Saed et al, 2021). MASH histologically manifests with micro- and macrosteatosis that is deposition of neutral lipid particles in the cytoplasm, accompanied by lobular inflammation, hepatocellular ballooning, Mallory-Denk bodies, nuclear atypia, and fibrosis (Gines et al, 2021; Kim et al, 2017). MASH frequently accompanies the intrahepatic cholestasis (Shipovskaya and Dudanova, 2018). Suppression of de novo triacylglycerol (TAG) biosynthesis in hepatocytes may ameliorate the MASH progression, while a common-use remedy of MASH from this point of view has yet to be established (Kim et al, 2017).

The rate of de novo TAG biosynthesis is limited by acetyl-CoA carboxylase 1 (ACC1) and pyruvate carboxylase (PC) in the liver (Bian et al, 2022; Kumashiro et al, 2013; Wakil and Abu-Elheiga, 2009). ACC1 carboxylates acetyl-CoA to produce malonyl-CoA, which facilitates fatty acid biosynthesis (Ye et al, 2019) and suppresses the mitochondrial β-oxidation enzyme, carnitine palmitoyltransferase (CPT1) activity (Folmes and Lopaschuk, 2007). On the other hand, PC carboxylates pyruvate to produce oxaloacetate, which enhances Krebs cycle and glycerol biosynthesis especially in the liver (Kumashiro et al, 2013). ACC1 degradation is promoted by constitutive photomorphogenic protein 1 (COP1), a ring-finger-like E3 ubiquitin ligase, which essentially regulates glucose and lipid metabolism (Ghosh et al, 2016; Kato et al, 2008). Clinically, ACC1 is regarded as the most promising target for MAFLD/MASH, while its better regulatory strategies are still needed to pursue for clinical intervention, because hyperlipidemia can be adversely caused by simple ACC1 inhibition strategy (Bian et al, 2022; Goedeke et al, 2018; Zhang et al, 2021).

We have approached this question through human and mouse molecular genetics of KIF12 kinesin. KIF12 is a member of kinesin-12 family of kinesin superfamily molecular motors (Hirokawa et al, 2009), and is ubiquitously expressed in internal organs (Yang et al,

[1]Department of Cell Biology and Anatomy, Graduate School of Medicine, The University of Tokyo, Hongo, Tokyo 113-0033, Japan. [2]Department of Genetics, Hadassah Hebrew University Medical Center, Jerusalem 91120, Israel. [3]Makassed Hospital, Faculty of Medicine, Al-Quds University, Jerusalem, Palestine. [4]Department of Advanced Morphological Imaging, Graduate School of Medicine, Juntendo University, 2-5-1, Hongo, Bunkyo-ku, Tokyo 113-8421, Japan. ✉E-mail: hirokawa@m.u-tokyo.ac.jp

2014). In the pancreatic islets, KIF12 suppresses oxidative stress and facilitates insulin secretion by scaffolding a nascent transcript factor Sp1 (Yang et al, 2014). In addition, human KIF12 mutations have been described in inherited inflammatory liver disease and cholestasis (Maddirevula et al, 2019; Stalke et al, 2021; Unlusoy Aksu et al, 2019). However, the relevance of KIF12 in lipid metabolism has totally been unidentified.

In the present study, we propose a KIF12-mediated MASH-preventing pathway through human and mouse genetic studies and cell biology. We identify KIF12 bi-allelic mutations in three inherited early-onset human liver cirrhosis pedigrees, one of which has been mimicked by mouse molecular genetics. Studies with gene silenced HepG2 cells have further revealed that KIF12 physiologically accelerates the turnover of ACC1 and PC, which may prevent the overproduction of TAG and disinhibits mitochondrial β-oxidation of fatty acids. This activity may rely on a non-motor scaffolding capacity of a KIF12-PRD, which can solely reduce the lipid droplet formation in general. In superresolution microscopy, we have visualized ternary interaction among microdroplets of KIF12, ACC1, and COP1, which may enhance the COP1 activity on facilitating the ubiquitination-mediated turnover of ACC1. Those findings will propose an anti-MASH pathway involving KIF12, which may significantly reduce the TAG contents in hepatocytes.

## Results

### KIF12 dysfunctional mutations in inherited human liver cirrhosis

In the course of whole exome sequencing of early-onset familial liver cirrhosis patients in Israel, we identified the following three human cases from consanguineous marriage carrying bi-allelic KIF12 mutations. The mutations were further confirmed by Sanger sequencing in patients' fibroblasts (Fig. 1A–C; Table 1).

Patient 1 was first diagnosed at 11 years old (y.o.). He suffered from a progressive portal hypertension manifested by hepatosplenomegaly (Fig. 1D), esophageal varices, recurrent ascites, deterioration in liver synthetic function, and died at 18. Liver biopsies revealed bridging fibrosis and neutrophil infiltration as the sign of liver cirrhosis (Fig. 1E). Micro- and macrovesicular steatosis and Mallory Denk Bodies (MDBs) were accompanied occasionally, but cholestasis was not histologically evident. He was homozygous for a rare variant in KIF12, giving rise to a p.Met338Thr (p.M338T) mutation (Fig. 1A–C; Table 1).

Patient 2 was diagnosed with liver dysfunction at the age of 2. She is currently 14 y.o. She is also suffering from hepatosplenomegaly (Fig. EV1A) and portal hypertension. She revealed pancytopenia, ascites, and esophageal varices, but without encephalopathy. The blood biochemistry revealed continuous high levels of aminotransferases including aspartate aminotransferase (AST; Fig. 1F) and alanine aminotransferase (ALT; Fig. 1G), suggesting the existence of severe hepatocellular damage. Simultaneously, we detected elevation of direct bilirubin (Fig. EV1B) and bile duct enzymes including alkaline phosphatase (ALP; Fig. 1H) and γ-GTP (Fig. EV1B), suggesting the existence of cholestasis. She is homozygous for another KIF12 rare variant, giving rise to p.Arg368Ter (p.R368X) nonsense mutation that may delete the whole non-motor domain of KIF12 (Fig. 1A–C).

Patient 3 (currently 20 y.o.) presented with liver dysfunction in infancy (Fig. EV1C). He also had continuous high levels of AST, ALT, and ALP (Fig. 1F–H). He had portal hypertension complicated by splenomegaly, thrombocytopenia, esophageal varices, and splenorenal shunt, without ascites or encephalopathy. In addition, he experienced multiple idiopathic syncope episodes in his life. He is homozygous for a rare KIF12 intronic variant in KIF12 c.1222+2 T > A that may disrupt a splicing donor sequence resulting in a frameshift mutation in the middle of the KIF12 protein (Fig. 1B), which was shared by his cousin who died in youth (Fig. EV1C).

The frequencies of these variants were very low in GnomAD database, and the pathogenicity of all variants were predicted to be significant by VarSome database (Kopanos et al, 2019). The M338T mutation in Patient 1 on a conserved methionine involves possible disruption of an essential β sheet of kinesin motor domain (Fig. 2A). The large truncations of whole non-motor domains in the C-terminus half (Patients 2 and 3) are predicted to also cause significant dysfunction of KIF12 protein. The C-term non-motor domain was considered to be most essential, because of the existence of KIF12 splicing variants without containing the whole or N-terminus of the kinesin motor domain (Fig. EV2A).

### KIF12 mutant mice suffered from MASH-like steatohepatitis

As KIF12 amino acid sequence is largely conserved between human and mouse (Figs. 2A and EV2A), we generated a mutant Kif12 mouse model by CRISPR/Cas9 technology (Tanaka et al, 2023) by mimicking the nonsense mutation p.R368X (human) carried by Patient 2 (Fig. 2B,C). As a result, 4 out of 96 F0 offspring were identified to contain the desired bi-allelic changes (Fig. EV2B,C). The KIF12 immunofluorescence signal in the liver of $Kif12^{Mut/Mut}$ mice was significantly reduced compared with that of control (Fig. EV2D,E).

Histologically, the H&E staining of $Kif12^{Mut/Mut}$ mouse liver showed MASH-like features including microsteatosis, Mallory-Denk bodies, and atypical nuclei in zone 3 area, indicating the occurrence of liver steatosis without apparent sign of cholestasis (4/4; Fig. 2D). Masson's trichrome staining showed perisinusoidal fibrosis in zone 3 (4/4). Consistent with these, the liver triacylglycerol (TAG) level of the $Kif12^{Mut/Mut}$ mouse liver was significantly elevated (Fig. 2E). These observations suggested that the mutation lead to early-onset MASH/MAFLD.

In blood biochemistry of four F0 homozygotes at 12 weeks old (w.o.), AST and ALT were elevated over several hundred IU/L (Fig. 2F,G). Lactate dehydrogenase (LDH) and direct bilirubin were also elevated (Fig. 2H). However, the values of γ-glutamyl transferase were below the detection limit of 9 IU/L in all examined animals ($n = 4$), indicating that sign of cholestasis was not very evident at this stage. Although the cholestasis according to KIF12 mutation can be progressive (Samanta et al, 2023), this KIF12 mutation from Patient 2 may monogenetically and primarily cause a MASH/MAFLD-like liver steatosis preceding the onset of apparent cholestasis.

### KIF12-PRD suppresses steatosis in hepatocytes

To investigate the KIF12 phenotype in vitro, we conducted KIF12 gene silencing in the human hepatocyte HepG2 cells. We

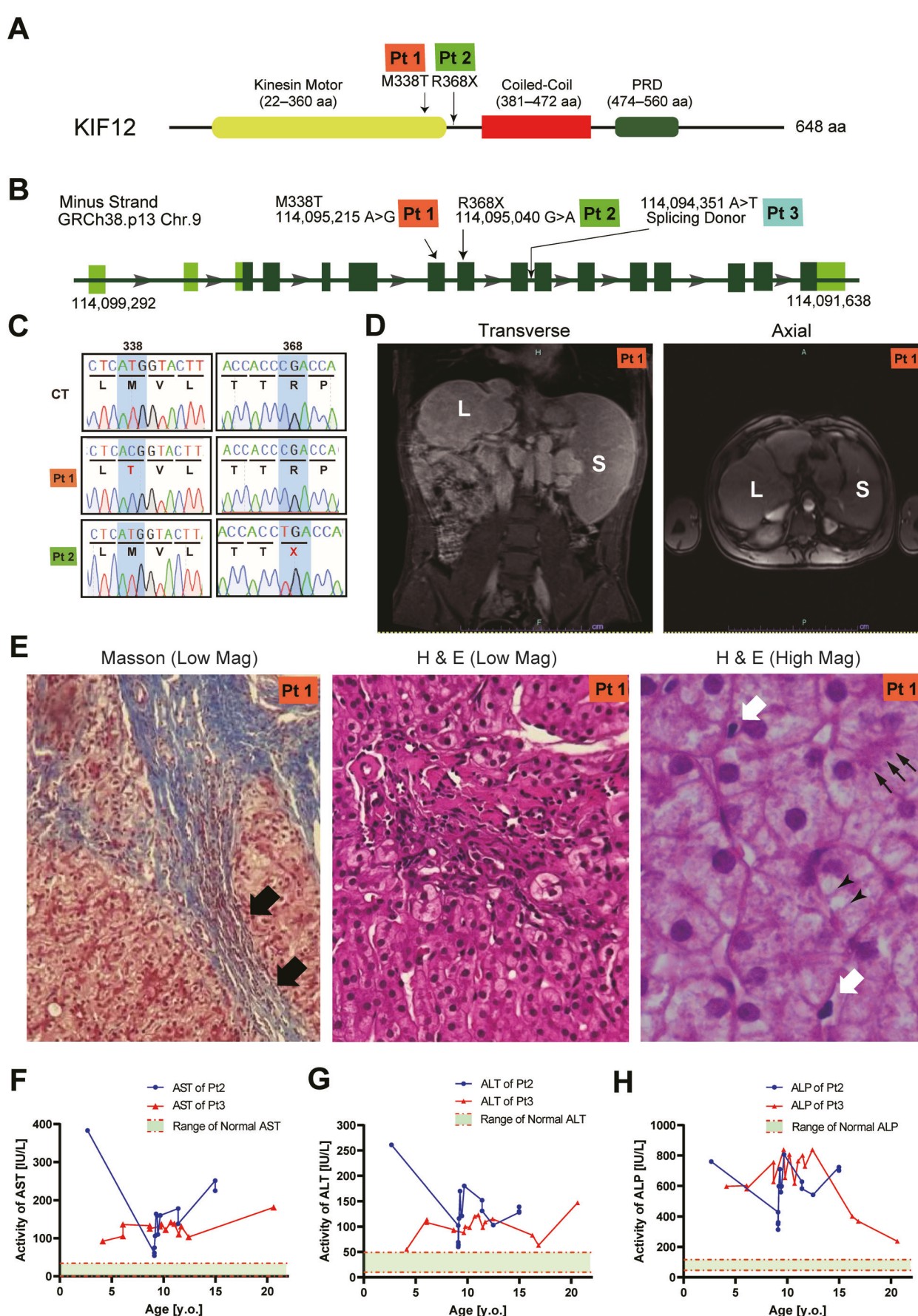

**Figure 1. Inherited KIF12 mutations in human liver cirrhosis patients.**

(A, B) Schematic representations of the human KIF12 mutations in cirrhotic patients 1–3 (Pt1–3) in the protein domain structure (A) and in exon-intron structure (B; NCBI Genome Assembly GRCh38, p13; Accession No. NM_138424). (C) Sanger sequence electropherograms of human fibroblast genomic DNAs of Patients 1 and 2 representing the bi-allelic mutated pairs giving rise to M338T (Pt1) and R368X (Pt2) mutations. (D) MRI images of Patient 1 indicating hepatosplenomegaly. Corresponds to Fig. EV1A. (E) Liver biopsy of Patient 1 with Masson's trichrome and Hematoxylin & Eosin (H&E) staining. Closed arrows, fibrosis. Open arrows, inflammatory cell infiltration. Arrowheads, steatosis. Small arrows, a Mallory-Denk body. (F–H) Clinical course of aspartate aminotransferase (AST; F), alanine aminotransferase (ALT; G), and alkaline phosphatase (ALP; H), in blood biochemistry of Patients 2 and 3. Corresponds to Fig. EV1B,C.

**Table 1. Summary of the liver cirrhosis patients with *KIF12* genetic mutations.**

| | Patient 1 | Patient 2 | Patient 3 |
|---|---|---|---|
| Sex | Male | Female | Male |
| Mutation (GRCh38, p13) | Chr9:114,095,215 A > G | Chr9:114,095,040 G > A | Chr9:114,094,351 A > T |
| Mutation (GRCh37, p13) | Chr9:116,857,495 A > G | Chr9:116,857,320 G > A | Chr9:116,856,631 A > T |
| dbSNPs | – | rs772936444 | rs1847120150 |
| Frequency | – | A = 0.000008 (2/242310, GnomAD_exome)<br>A = 0.000014 (2/140212, GnomAD) | T = 0.000007 (1/140110, GnomAD) |
| Expected amino acid change (NM_138424) | *p*.Met338Thr | *p*.Arg368Ter | *c*.1222+2 T > A<br>Splicing donor mutation |
| Age diagnosed with the disease | 11 y.o.–died at 18 y.o. | 14 y.o. | 20 y.o. |
| Family history/consanguinity | • Parents are first cousins. | • Parents are first cousins.<br>• A brother died of liver disease. | • Parents are first degree cousins.<br>• A similarly affected cousin died and later found to be homozygous for the same mutation. |
| Liver function tests in plasma | • Elevated bilirubin<br>• Normal albumin<br>• Elevated ALT, AST | • Elevated bilirubin<br>• Elevated ALT, AST<br>• Normal α-fetoprotein<br>• Normal albumin | • Elevated bilirubin<br>• Elevated ALT, AST<br>• Normal α-fetoprotein<br>• Low albumin |
| Liver-related sonographic and endoscopic findings | • Hepatosplenomegaly<br>• Portal hypertension<br>• No portal thrombosis<br>• Esophageal varices | • Hepatosplenomegaly<br>• Portal hypertension<br>• Esophageal varices<br>• No portal thrombosis<br>• No hepatic occlusion | • Hepatosplenomegaly<br>• Portal hypertension<br>• Esophageal varices<br>• Spontaneous splenorenal shunt<br>• No hepatic occlusion<br>• No portal thrombosis |
| Liver histological findings | • Neutrophil infiltration<br>• Bridging fibrosis<br>• Heterogenous liver | • Hepatic fibrosis in infancy<br>• Cholestasis<br>• Bridging fibrosis<br>• Periportal globules, PAS-D positive | • Hepatic fibrosis in infancy<br>• Cholestasis<br>• Bridging fibrosis |

transduced RNA-polymerase-II-based miRNA expression vectors of either KIF12-antisense knockdown (KD) or scrambled control (SC) sequences (Yang et al, 2014). This SC sequence was designed to be apart from any known human cDNAs, so that it was served to exclude any general artifact of miRNA transduction. According to immunocytochemistry (Fig. 3A–C) and immunoblotting (Fig. 3D), a significant reduction in the KIF12 level was observed in the KD cells. Neutral lipid particles were significantly accumulated in their cell bodies, which were restored by RNAi-immune mCitrine-KIF12 expression (Fig. 3E–G).

To identify the responsible domain of KIF12 against this phenotype, we generated a series of truncated RNAi-immune mouse *Kif12* cDNAs, which were, respectively, tagged with mCitrine (mCit; Fig. 3H,I). They were co-transduced with the KD vector into HepG2 cells, and neutral lipid staining was conducted (Fig. 3J–N). Interestingly, constructs carrying the KIF12-PRD, i.e., mCit-KIF12-PRD and mCit-KIF12-Tail significantly reduced the neutral lipid levels in KIF12-KD cells, even below that of the SC cells transduced with mCit alone as a normal

control (SC + NC). mCit-KIF12-Head or mCit-KIF12-Stalk scarcely reduced it, suggesting a dominant suppressing activity of KIF12-PRD against lipid accumulation.

We further tested if KIF12-PRD can generally ameliorate lipid particle accumulations in fatty-acid-loaded hepatocytes (Fig. EV3A,B) (Tsai et al, 2007). Fatty acid loading enhanced the lipid droplet accumulation. mCit-KIF12-PRD transduction for 48 h significantly reduced it compared with mCit-alone transduction. These data suggested that KIF12-PRD can suppress the progression of steatosis caused by overnutrition.

In the meanwhile, we conducted transdifferentiation of human fibroblasts into hepatocyte-like hiHep cells by introducing a set of transcription factors (Fig. EV3C) (Huang et al, 2014). Elevation of Hepatocyte-Specific Antigen (HSA) expression ensured successful transdifferentiation into hepatocyte-like cells (Fig. EV3D). Neutral lipid levels of hiHep cells derived from Patients 1 and 2 were significantly higher than those from control (CT) subjects (Fig. EV3E,F). KIF12-PRD transduction significantly reduced the neutral lipid levels of patients' hiHep cells. These data suggested

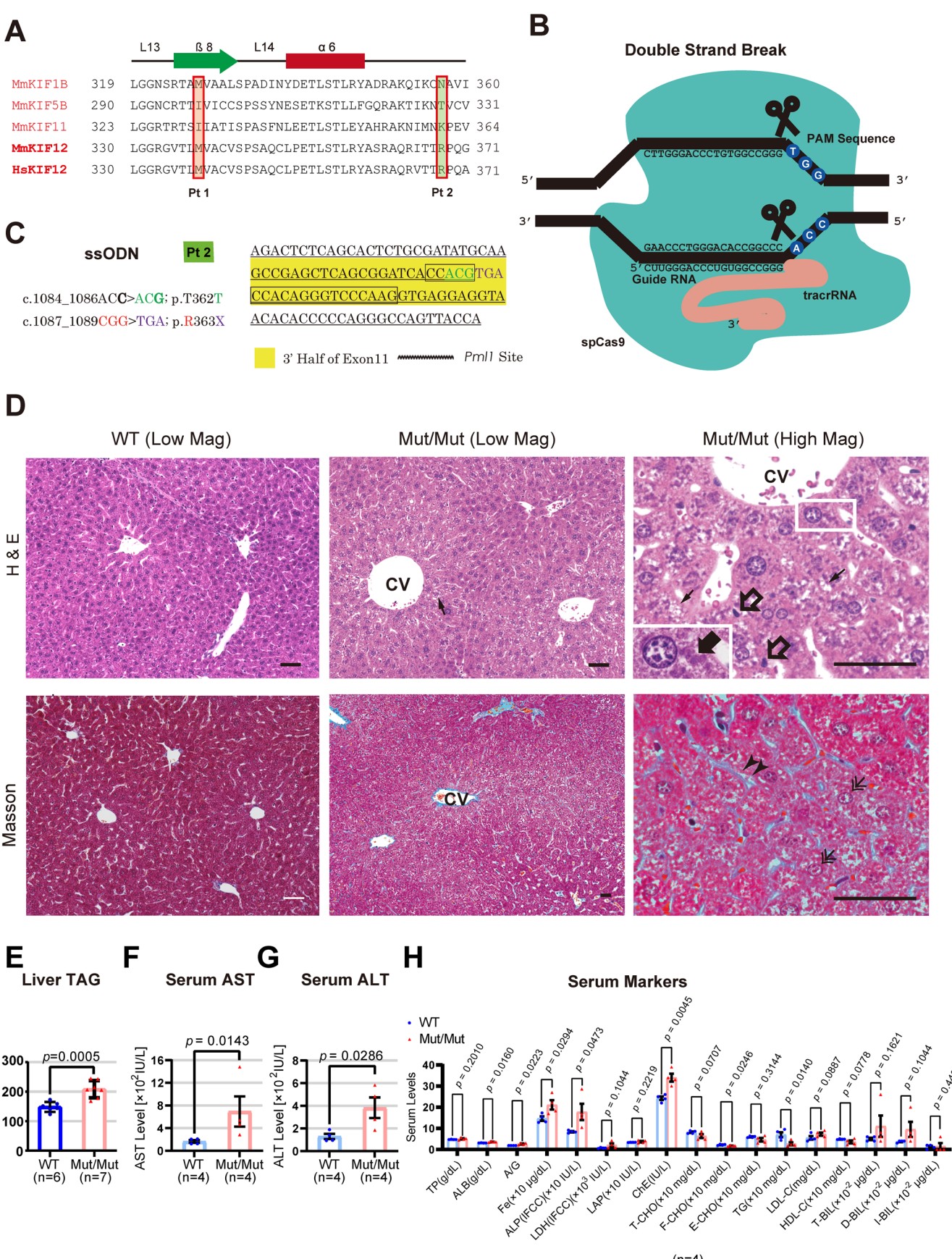

**A**

L13  ß 8  L14  α 6

| MmKIF1B | 319 | LGGNSRTA**M**VAALSPADINYDETLSTLRYADRAKQIKC**N**AVI | 360 |
| MmKIF5B | 290 | LGGNCRTTI**I**VICCSPSSYNESETKSTLLFGQRAKTIKN**T**VCV | 331 |
| MmKIF11 | 323 | LGGRTRTSI**I**ATISPASFNLEETLSTLEYAHRAKNIMNK**P**EV | 364 |
| **MmKIF12** | 330 | LGGRGVTI**M**VACVSPSAQCLPETLSTLRYASRAQRITT**R**PQG | 371 |
| **HsKIF12** | 330 | LGGRGVTI**M**VACVSPSAQCLPETLSTLRYASRAQRVTT**R**PQA | 371 |

Pt 1                                Pt 2

**B**

**Double Strand Break**

PAM Sequence

CTTGGGACCCTGTGGCCGGG  T G G

5′                                                    3′

3′                                                    5′

GAACCCTGGGACACCGGCCC  A C C

5′ CUUGGGACCCUGUGGCCGGG

Guide RNA                              tracrRNA

spCas9                            3′

**C**

**ssODN**    [Pt 2]

c.1084_1086AC**C**>AC**G**; p.T362T
c.1087_1089**CGG**>**TGA**; p.R363**X**

AGACTCTCAGCACTCTGCGATATGCAA
GCCGAGCTCAGCGGATCACC**ACG**TGA
CC**ACAGGGTCCCAAG**GTGAGGAGGTA
ACACACCCCCAGGGCCAGTTACCA

🟨 3' Half of Exon11    〰〰 *Pml*1 Site

**D**

|   | WT (Low Mag) | Mut/Mut (Low Mag) | Mut/Mut (High Mag) |

H & E

Masson

**E**  Liver TAG
p=0.0005
Lipid Concentration [mg/dL]
WT (n=6)  Mut/Mut (n=7)

**F**  Serum AST
p = 0.0143
AST Level [×10² IU/L]
WT (n=4)  Mut/Mut (n=4)

**G**  Serum ALT
p = 0.0286
ALT Level [×10² IU/L]
WT (n=4)  Mut/Mut (n=4)

**H**  Serum Markers
● WT  ▲ Mut/Mut
Serum Levels

p=0.2010, p=0.0160, p=0.0223, p=0.0294, p=0.0473, p=0.1044, p=0.2219, p=0.0045, p=0.0707, p=0.0246, p=0.3144, p=0.0140, p=0.0887, p=0.0778, p=0.1621, p=0.1044, p=0.4408

TP(g/dL), ALB(g/dL), A/G, Fe(×10 μg/dL), ALP(IFCC)(×10 IU/L), LDH(IFCC)(×10³ IU/L), LAP(×10 IU/L), ChE(IU/L), T-CHO(×10 mg/dL), F-CHO(×10 mg/dL), E-CHO(×10 mg/dL), TG(×10 mg/dL), LDL-C(mg/dL), HDL-C(×10² mg/dL), T-BIL(×10² μg/dL), D-BIL(×10² μg/dL), I-BIL(×10² μg/dL)

(n=4)

◄ **Figure 2. A mouse model mimicking a human KIF12 truncation mutation reproduced MASH-like phenotypes.**

(A) Amino acid sequence alignment among various human kinesins and mouse KIF12 proteins. L, loop. α, alpha helix. β, beta-sheet. Note that the mutated residues are conserved between human and mouse KIF12 proteins. Corresponds to Fig. EV2A. (B, C) Mouse gene editing strategy by CRISPR/Cas9 technology, which breaks the double strand by the guide RNA (B) and introduces a KIF12 nonsense mutation R363X by ssODN (C). Corresponds to Fig. EV2B–E. (D) Histopathology of wild type (WT) and *Kif12^{Mut/Mut}* #7 mouse liver at 12-week-old with H&E staining and Masson's trichrome staining. CV, central vein. Small arrows, microsteatosis; open arrows, inflammatory cell infiltration; closed arrow in the inset, Mallory-Denk body; double arrows, atypical nuclei with condensed nucleolus; arrowheads, perisinusoidal fibrosis. Scale bars, 50 μm. (E) Liver TAG levels. One-sided Welch's *t* test. Biological replicates, individual animals. (F–H) Summary of serum biochemistry examinations of transaminases (F, G) and other markers (H). Error bars represent mean ± SEM. One-sided Mann-Whittney's test (E, F) and Welch's *t* test (G). Biological replicates, mouse individuals.

that Patients 1 and 2 likely primarily suffered from liver steatosis, essentially because of the loss of KIF12 PRD expression.

We compared the shapes of mitochondria and lysosomes on KIF12-PRD overexpressing HepG2 cells (Fig. EV4) to exclude significant toxicity of this polypeptide on the cells. As a result, mitochondrial complexity was likely to increase in the over-expressing cells, while the size and number of lysosomes appeared without changes. This mitochondrial expansion may reflect the upregulated mitochondrial β oxidation, as assessed in later analysis.

## KIF12-PRD binds to ACC1 and PC

To investigate the underlying molecular mechanism of KIF12-PRD-mediated prevention of steatosis, binding partners of KIF12-PRD were screened by a BioID proximity biotinylation assay in HepG2 cells. Using mass spectrometry and immunoblotting, we identified the lipid biosynthesis enzymes, ACC1 and PC, as candidates of the binding partners (Fig. 4A). A GST pulldown assay with mouse liver lysates revealed significantly stronger association of KIF12-PRD-GST with ACC1 than that of GST alone (Fig. 4B). A tag-proximity ligation assay (Tag-PLA) in HepG2 cells (Fig. 4C–E), mCit-KIF12-FL and mCit-KIF12-PRD were associated with ACC1 and PC significantly stronger than mCit alone. These data proposed a specific binding capacity of KIF12-PRD toward ACC1 and PC, consistent with the intrinsic colocalization among the microdroplets by superresolution immunocytochemistry as shown later.

## ACC1 and PC levels were elevated by KIF12 deficiency

Interestingly, KIF12 deficiency led to upregulation of ACC1 and PC in the liver (Fig. 5A–D). Immunohistochemistry of *Kif12^{Mut/Mut}* mouse livers at 12 w.o. revealed significantly higher levels in ACC1 and PC expression than control (Fig. 5A,C). Immunocytochemistry of KIF12-KD cells also revealed upregulation of ACC1 and PC signals compared with the scrambled control (SC) cells (Fig. 5B,D). Their expression in cell periphery was abnormally increased. This upregulation in ACC1 level was reproduced by immunoblotting of KIF12-deficient HepG2 cell lysates (Fig. 5E,F). These data suggested that KIF12 serves as a negative regulator of ACC1 and PC protein expression.

Because the product of ACC1 enzyme, malonyl-CoA, is known to inhibit mitochondrial carnitine palmitoyltransferase (CPT1) activity (Folmes and Lopaschuk, 2007), we sought to investigate if *Kif12^{Mut/Mut}* liver is also impaired in fatty acid metabolism. We biochemically measured the CPT1 activity of wild type and *Kif12^{Mut/Mut}* livers of 8 weeks old. As a result, the CPT1 activity of mutant mouse livers were significantly reduced (Fig. 5G). This suggests that KIF12 not only suppresses de novo neutral lipid

biosynthesis but also augments the fatty acid oxidation in the liver (Fig. 5H), as its relevant mechanism against liver steatosis.

To examine whether the ACC1 and/or PC upregulation caused the lipid droplet accumulation in KIF12 deficient cells, we conducted the following three experiments (Fig. EV5). First, overexpression of these proteins similarly led to significant lipid droplet accumulation in HepG2 cells (Fig. EV5A–C,F,G). Then, we treated the cells with the ACC1 inhibitor, TOFA (5-tetradecyloxy-2-furoic acid). The lipid droplet levels of KIF12-KD cells were reduced to the control level (Fig. EV5D,H), suggesting that ACC1 could be a relevant effector of KIF12-mediated anti-steatosis pathway. We also suppressed the ACC1 expression by shRNA transfection. This experiment provided a consistent result (Fig. EV5E,I). These data collectively supported the causal relationship of KIF12-mediated suppression of ACC1 and/or PC levels that significantly prevented steatosis in hepatocytes.

## KIF12 accelerates ACC1 ubiquitination and turnover

We conducted qRT-PCR to measure the *ACC1* and *PC* mRNA levels (Fig. 6A). They paradoxically tended to decrease in KD cells, being opposite to the tendency in the protein expression levels. This discrepancy may be probably due to a negative feedback mechanism according to upregulated protein expression. Actually, enzymatic product acyl-CoA is reported to negatively regulate ACC1 transcription (Faergeman and Knudsen, 1997). Thus, the increase in ACC1 and PC protein expression levels may be mainly caused by a posttranscriptional change due to KIF12 deficiency.

We then performed cycloheximide (CHX)-mediated protein turnover assays in KIF12-KD and SC cells to assess the molecular mechanism of the upregulation (Fig. 6B–E). KIF12-deficiency significantly reduced the initial turnover rates of ACC1 and PC proteins, approximately to half. Interestingly, KIF12 protein turnover also occurred in a higher rate, which may explain the gradual reduction of the ACC1 and PC turnover rates in SC cells after 2 h of treatment.

As ubiquitin-proteasomal system (UPS) is one of the major pathways of protein turnover (Piper et al, 2014), we treated the KIF12-KD system by proteasomal and lysosomal inhibitors, MG132 and chloroquine, respectively (Fig. 6F,G). The levels of ACC1 in SC and KD cells became similar after the inhibitor treatment, suggesting that the difference in degradation rates was a cause of the discrepancy in the ACC1 expression levels. On this occasion, we conducted ACC1 immunoprecipitation to assess its ubiquitination levels among the KIF12-KD system. The multi-ubiquitin levels of KD cell precipitants were significantly lower than those of SC cell precipitants. These data suggested that KIF12 is essentially involved in ACC1 ubiquitination for its turnover.

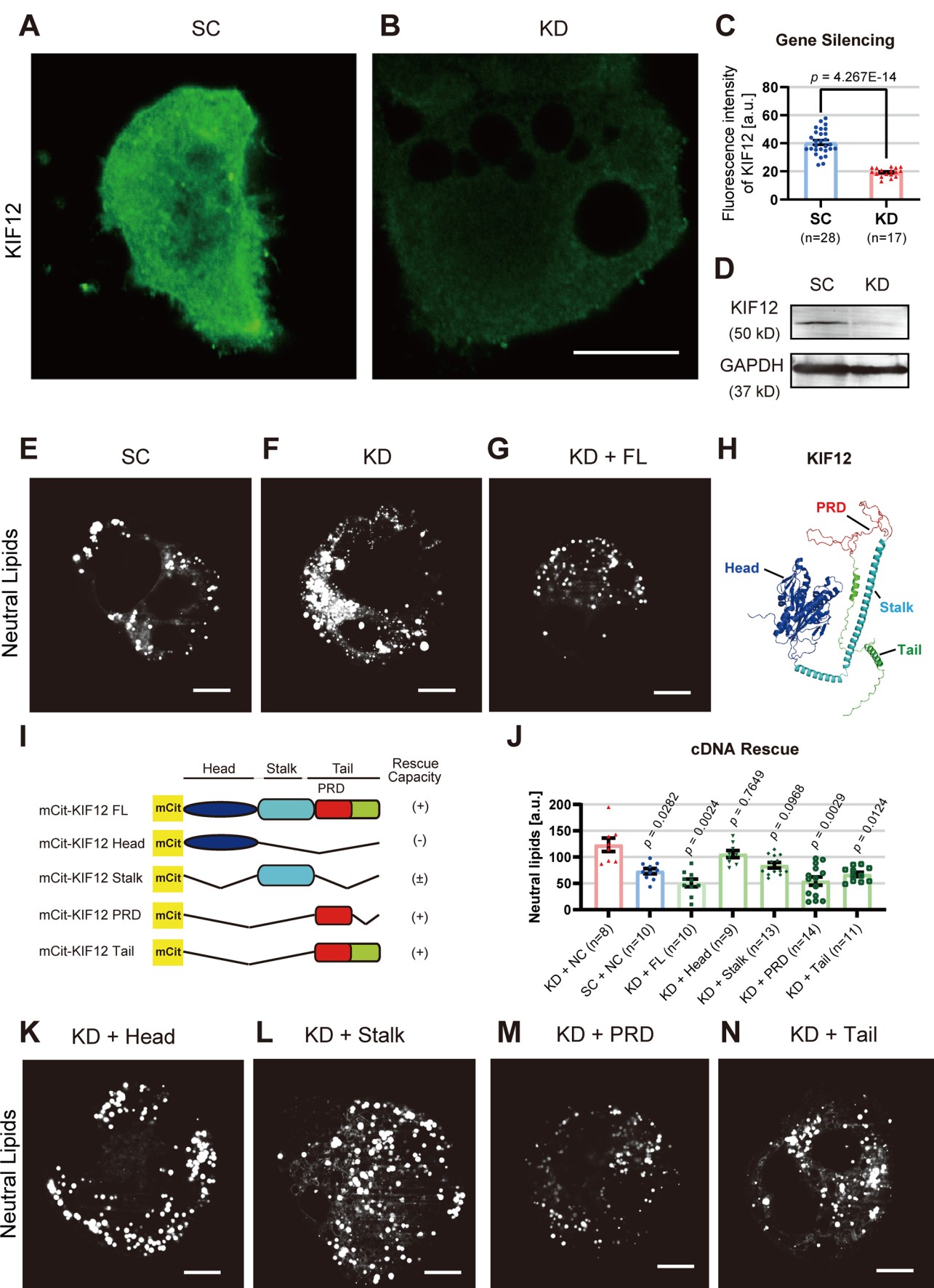

**A** SC KIF12

**B** KD

**C** Gene Silencing
*p* = 4.267E-14
Fluorescence intensity of KIF12 [a.u.]
SC (n=28)    KD (n=17)

**D**
SC    KD
KIF12 (50 kD)
GAPDH (37 kD)

**E** SC — Neutral Lipids

**F** KD

**G** KD + FL

**H** KIF12
PRD
Head
Stalk
Tail

**I**
| | Head | Stalk | Tail PRD | Rescue Capacity |
|---|---|---|---|---|
| mCit-KIF12 FL | mCit | | | (+) |
| mCit-KIF12 Head | mCit | | | (−) |
| mCit-KIF12 Stalk | mCit | | | (±) |
| mCit-KIF12 PRD | mCit | | | (+) |
| mCit-KIF12 Tail | mCit | | | (+) |

**J** cDNA Rescue
Neutral lipids [a.u.]
KD + NC (n=8)
SC + NC (n=10)    *p* = 0.0282
KD + FL (n=10)    *p* = 0.0024
KD + Head (n=9)    *p* = 0.7649
KD + Stalk (n=13)    *p* = 0.0968
KD + PRD (n=14)    *p* = 0.0029
KD + Tail (n=11)    *p* = 0.0124

**K** KD + Head — Neutral Lipids

**L** KD + Stalk

**M** KD + PRD

**N** KD + Tail

**Figure 3.  KIF12-PRD is essential and sufficient against lipid droplet accumulation.**

(A–D) Characterization of KIF12 gene silencing in HepG2 cells, represented by KIF12 immunofluorescence microscopy of those transduced by scrambled control (SC; **A**) and knockdown (KD) miRNAs using an anti-KIF12 antibody #SC-48558 (**B**), accompanied by its statistics (**C**); and immunoblotting using an anti-KIF12 antibody #E28700-C which was thrice reproduced with separate dishes (**D**). Scale bars, 20 μm. Error bars, mean ± SEM. Welch's *t* test. Biological replicates, individual cells. (E–G) LipidTOX neutral lipid staining of HepG2 cells transduced by SC miRNA (**E**), KD miRNA (**F**), and KD miRNA plus mCitrine (mCit)-tagged RNAi-immune full-length mouse *Kif12* cDNA (**G**). Scale bars, 10 μm. (H) Alpha Fold prediction of the human KIF12 protein structure. Note that KIF12-PRD constitutes a nonstructured domain on the top right. Reproduced from https://www.uniprot.org/uniprotkb/Q96FN5/entry. (I–N) Phenotypic rescue experiments using the mCit-tagged *Kif12* cDNA constructs (**I**), quantified by the neutral lipid levels (**J**) according to LipidTOX staining of the cells after transduced by the indicated vectors (**K–N**). Scale bars, 10 μm. Error bars, mean ± SEM. One-way ANOVA against KD + NC (mCit alone). Biological replicates, individual cells. Corresponding to Figs. EV3 and EV4.

To investigate the molecular mechanism of ACC1 ubiquitination, we conducted shRNA-mediated silencing of *COP1* gene, which was previously described as the responsible E3 ubiquitin ligase for ACC1 (Ghosh et al, 2016; Kato et al, 2008). We suppressed ACC1 expression level by overexpressing KIF12-PRD in HepG2 cells. In this condition, COP1 shRNA co-transduction significantly elevated the ACC1 immunofluorescence (Fig. EV6A,B).

Then, we further compared the binding capacity of ACC1 to COP1, with co-immunoprecipitation analyses. The ACC1–COP1 coprecipitating capacity was significantly reduced by the KIF12 deficiency in a reproducible manner (Fig. 6H,I), suggesting a scaffolding role of KIF12 in facilitating ACC1 ubiquitination by the E3 ligase, COP1.

## KIF12 makes a ternary complex with ACC1 and COP1

The above data suggested a scaffolding function of KIF12-PRD for ACC1 ubiquitination by COP1. Accordingly, we investigated the nano-scale relationship among KIF12, ACC1, and COP1 proteins using superresolution immunocytochemistry with a lattice-SIM microscope, Elyra7. To our surprise, we observed multiple triple-stained punctata in the cytoplasm, which may reveal the morphological nanostructures of so-called 'soluble' proteins in the cytoplasm (Fig. 7A,B).

In high magnification views at a sub-100 nm resolution, KIF12-containing structures frequently bridged ACC1- and COP1-containing microdroplets in a reproducible manner (Fig. 7C,D). To assess the colocalization tendency among those three proteins, we changed the images into binary and measured the occupying areas of each color and each overlapping pair of colors (Fig. 7E). Then we calculated the Colocalization Factors (CFs) in 5 independent cells (Fig. 7F). The triple colocalization area was 5–10 times larger than its expected area, suggesting that this triple colocalization is significant. Furthermore, the area with KIF12–ACC1 colocalization but without COP1; and that with KIF12–COP1 colocalization but without ACC1 were significantly larger than that with ACC1–COP1 colocalization but without KIF12. These results suggested a molecular mechanism that KIF12 scaffolds between ACC1 and COP1 microdroplets to facilitate the ACC1 protein turnover (Fig. 7G). We consider that this unique capacity of KIF12 may underlie its monogenic relevance against MASH progression.

## Overnutrition tends to decrease KIF12 protein expression

To investigate the relationship between metabolic syndrome and KIF12-mediated liver pathogenesis, we loaded oleic acid to HepG2 cells and assessed the possible changes in KIF12 expression both at protein and mRNA levels. Immunofluorescence microscopy revealed that the oleic acid loading significantly reduced the

KIF12 protein expression in HepG2 cells (Fig. 8A–C). In a basal state, KIF12 provided cytoplasmic distribution with peripheral dominance. After oleic acid treatment, the staining of KIF12 especially in the cell periphery was significantly reduced. These results suggested that overnutrition can downregulate KIF12.

Interestingly, the amount of *KIF12* mRNA was paradoxically elevated by the oleic acid treatment (Fig. 8D). This was also largely consistent with previous transcriptome studies on rodent MASH/MAFLD models (Figs. 8E,F and EV7). The reason of this discrepancy between the protein and mRNA levels is elusive, while the high rate of KIF12 protein turnover (Fig. 6E) implied that an unknown modulation mechanism of KIF12 protein turnover is involved in this overnutrition-induced reduction of KIF12 protein.

# Discussion

In this study, we have uncovered a relevant anti-MASH pathway in human and mouse scaffolded by KIF12 kinesin. The human patients and the mouse model lacking KIF12 C-terminal domain consistently suffer from MASH-like symptoms. We have shown that KIF12-PRD within this C-terminal domain directly accelerates the turnover of ACC1 and PC and prevents steatosis in hepatocytes. This will be one of the first indications on KIF12's monogenic relevance on early-onset human MASH pathogenesis, according to a unique molecular mechanism. Because KIF12 protein tended to decrease by oleic acid loading to HepG2 cells (Fig. 8), this mechanism may be partly involved in overnutrition-based MASH/MAFLD pathogenesis.

In previous studies, the human liver disease accompanying KIF12 mutations have been simply described as 'cholestasis' (Maddirevula et al, 2019; Stalke et al, 2021; Unlusoy Aksu et al, 2019). However, our detailed investigations in KIF12-defective human and mouse revealed that MASH should also occur in the upstream or in parallel to the development of cholestasis. Although high levels of ALP and direct bilirubin were detected in the sera (Figs. 1H, 2H and EV1B), the cholestasis was not histologically evident either in human, or in mouse (Figs. 1E and 2D). The mice did not elevate blood γ-GTP levels at 12 w.o., while MASH-like features such as microsteatosis, Malory-Denk bodies, atypical nuclei, and liver inflammations were already evident in histology (Fig. 2D). This was consistent with the human histopathology of Patient 1 (Fig. 1E). In addition, KIF12-deficient HepG2 cells revealed significant steatosis (Fig. 3). These findings consistently suggested that KIF12 deficiency may primarily cause steatosis in hepatocytes, before cholestasis becomes histologically evident.

As a molecular mechanism, the fatty acid synthesizing enzymes, ACC1 and PC, were significantly upregulated in KIF12-deficient cells

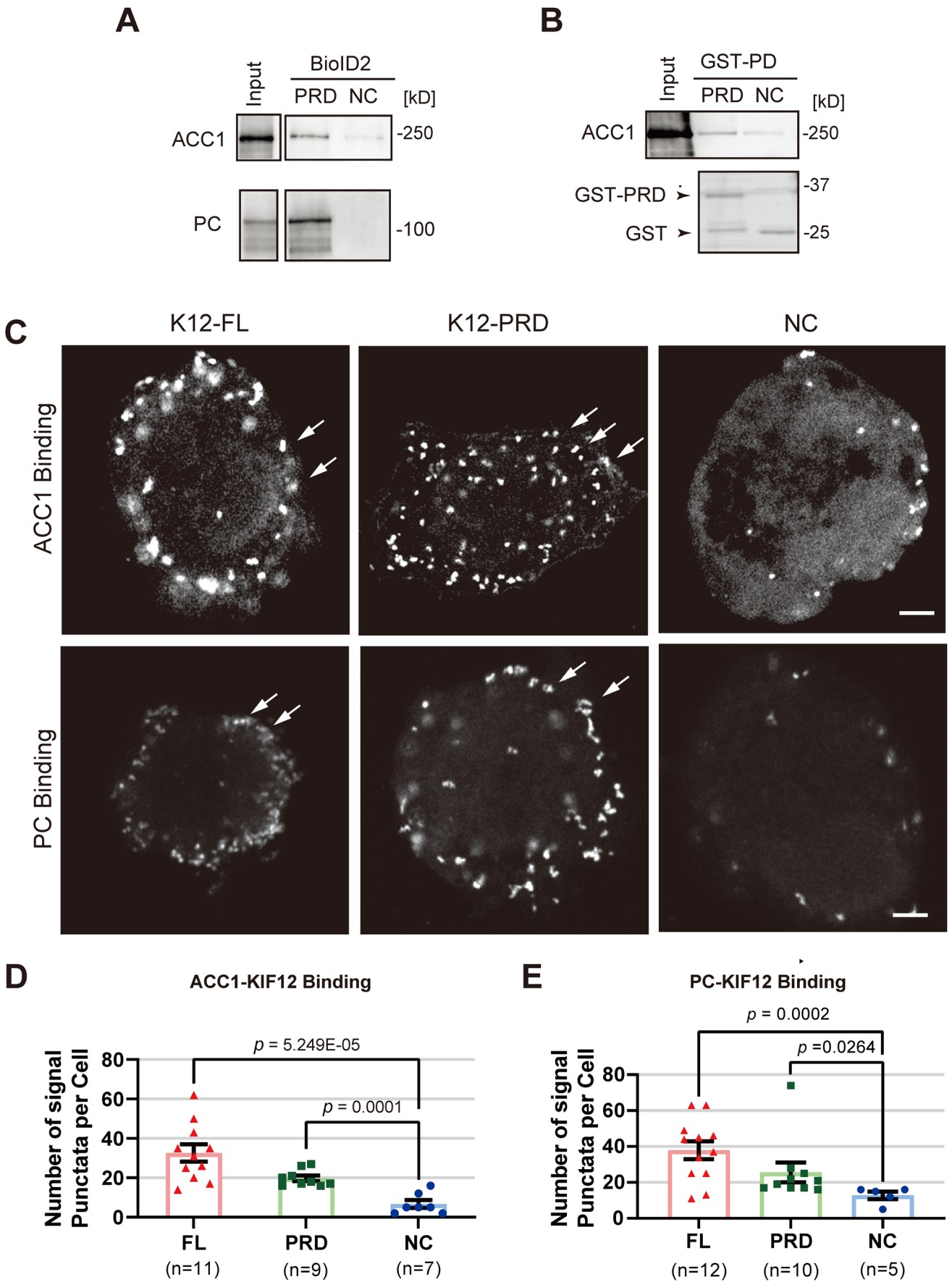

◄

**Figure 4.  KIF12-PRD is associated with ACC1 and PC proteins.**

(A) Immunoblotting of BioID assay precipitates in HepG2 cells stably transduced with KIF12-PRD-BioID2 (PRD) and BioID2 alone (NC) constructs. (B) GST pulldown assay of mouse liver lysates (Input) against KIF12-PRD-GST (PRD) and GST-alone (NC), shown by ACC1 immunoblotting (upper) and CBB staining (lower). Note that GST-PRD specifically pulled down ACC1. (C–E) Fluorescence images of a tag-proximity ligation assay in HepG2 cells being transduced with mCit-KIF12-FL, mCit-KIF12-PRD, and mCit alone (NC), respectively, which were proximity-labeled by an anti-GFP antibody and anti-ACC1 or anti-PC antibodies (C), with statistics (D and E). Scale bars, 10 μm. Error bars, mean ± SEM. Welch's *t* test. Biological replicates, individual cells. Arrows, protein interaction signals.

(Fig. 5). As overexpression of ACC1 or PC caused steatosis in wild-type hepatocytes (Fig. EV5A–C) and that the ACC1 inhibitor TOFA or the ACC1 shRNA ameliorated the steatosis in KIF12-deficient hepatocytes (Fig. EV5D,E), the causal relationship between the upregulation of those enzymes and the MASH pathology appeared to be very evident. We detected that KIF12 directly binds to and accelerates the ubiquitination of ACC1 by COP1 that may control the fatty acid biosynthesis (Figs. 4, 6F–I, 7 and EV6). Because KIF12 also scaffolds nascent transcription factor Sp1 until it is stabilized by posttranslational modifications in pancreatic beta cells (Yang et al, 2014), KIF12 may commonly enhance such protein modifications in various systems. This scaffolding capacity of KIF12 may be apart from the kinesin motor activity, because the motor-less KIF12-PRD overexpression can still ameliorate the steatosis either in KIF12 KD cells, in lipid-treated cells, or in human patient hiHep cells (Figs. 3 and EV3). Some kinesins have been reported to serve for scaffolding functions that can alter the geometrical and functional properties of the cargos, irrespective of their motility (Morikawa et al, 2018; Wang et al, 2018; Yang et al, 2014). Consistently, some KIF12 splicing variants lacked the whole or a part of the kinesin motor domain according to in silico research (Fig. EV2A) (Katoh and Katoh, 2005). The variant NP_612433.1 lacked N-terminal two-thirds of the motor domain that is dedicated to the mechanochemical ATPase cycle (Nitta and Hirokawa, 2020), but only leaving the C-terminal microtubule-binding α-helix domain. Thus, this KIF12-variant protein species is expected to preserve the microtubule-binding capacity but defective in motility, which will be subjected to future research. The existence of such a splicing variant supported our working hypothesis that KIF12 mainly serves as a molecular scaffold in hepatocytes.

As a morphological feature of this scaffolding complex, our lattice-SIM superresolution microscopy revealed that the COP1–KIF12–ACC1 ubiquitination machinery is likely composed of 100-nm-scale micro-droplets of the respective molecules (Fig. 7). This observation was largely consistent with our previous PALM/STORM data of the Sp1–KIF12 complex in pancreatic beta cells (Yang et al, 2014). It is much astonishing that the E3 ligase COP1 and their substrates can constitute such nano-scale non-membranous organelle complexes for accomplishing the enzymatic reactions. Those microdroplets may be formed through nano-scale liquid-liquid phase separation (LLPS) of 'soluble' proteins (Naz et al, 2024). How kinesins can modulate the behaviors of those microdroplets along microtubules will be an exciting issue of kinesin biology in the next generation. Further integration of nano-scale microscopy with molecular genetics will provide deeper insights on the nano-scale interaction underlying KIF12-mediated prevention of MASH/MAFLD-like pathogenesis.

These data will also illuminate the importance of KIF12-mediated pathway in translational research against MASH. Actually, there have been revealed several adverse effects of anti-MASH remedies. Hyperlipidemia has been accounted for the major adverse effect of conventional ACC1 inhibitor treatment (Bian et al,

2022; Goedeke et al, 2018; Kim et al, 2017). The ACC1 destabilizer IMA-1 better ameliorated the steatosis without causing hyperlipi-demia (Zhang et al, 2021). However, it simultaneously antagonized the p53-mediated tumor-suppressing pathway, which may raise the risk of carcinogenesis. Newly-developed thyroid hormone receptor beta (THR-β)-selective agonist Resmetirom nicely suppressed human liver fibrosis in 25% of MASH patients, while it caused severe diarrhea and nausea in 10% of patients (Harrison et al, 2024). On the other hand, we detected that KIF12 dually suppresses the expression of ACC1 and PC (Fig. 5), which serve as rate-limiting enzymes for fatty acid and glycerol biosynthesis, respectively, in the liver (Jitrapakdee et al, 2006; Kumashiro et al, 2013; Mao et al, 2006). Furthermore, we detected that KIF12 deficiency impairs the activity of liver mitochondrial β-oxidation enzyme, CPT1 (Fig. 5G,H). Those multiple enzyme regulation mechanism is thus expected to more thoroughly suppress the neutral lipid biosynthesis than simply inhibiting ACC1 and better circumvent the hyperlipidemia caused by ACC1 inhibition. However, because the involvement of inflammatory cells is also essential for MASH/MAFLD progression (Huby and Gautier, 2022), the actual efficacy of KIF12-mediated pathway in MASH/MAFLD therapeutics should be tested by future preclinical studies.

Accordingly, this study develops a cross-discipline strategy for investigating a clinically relevant scaffolding activity of KIF12 kinesin. It provides deeper understanding of a kinesin-mediated nano-scale regulation among the 'soluble' lipid biosynthesis enzymes, which now turns out to be essential for maintaining their own turnover rates. It accordingly prevents the occurrence of MASH as evidenced by human and mouse molecular genetics. The discovery of this nano-scale regulation mediated by KIF12 kinesin will therefore accelerate the elucidation of anti-MASH molecular machinery, and stimulate next-generation translational research into kinesins on the regulation of refractory metabolic diseases.

## Methods

### Human subjects

All patients and/or their parents provided written informed consent. MRI, blood biochemistry analysis, histopathology, and skin biopsies were performed with standard methods. Whole exome analysis was performed and interpreted as previously described (Ta-Shma et al, 2017). The patients are summarized in Table 1.

### Knockin mouse

Human-mutation-R363X-mimicking *Kif12*[Mut/Mut] mice were generated using CRISPR/Cas9 technology at Transborder Medical Research Center, University of Tsukuba, basically using previously described

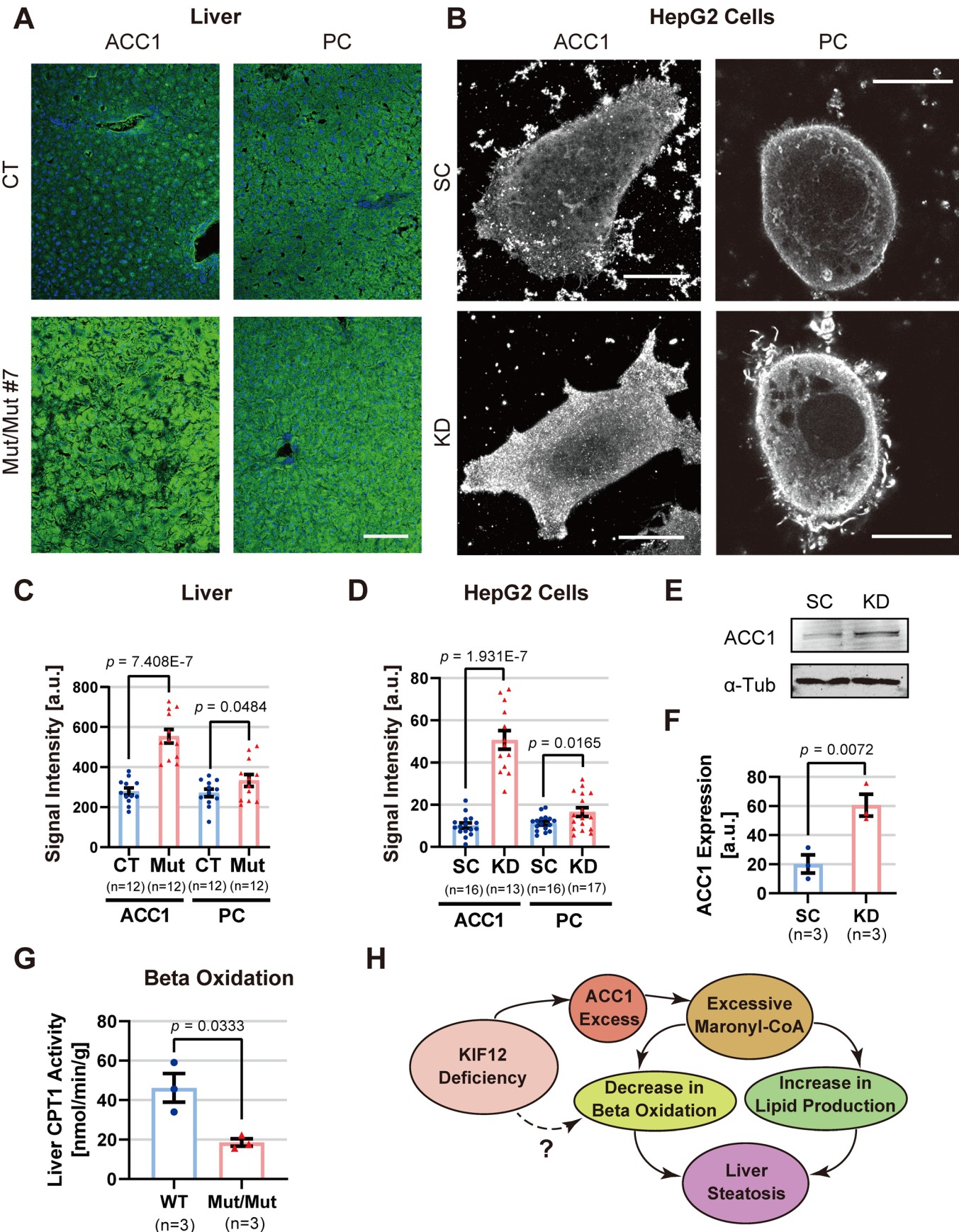

**A** Liver

ACC1 PC

CT

Mut/Mut #7

**B** HepG2 Cells

ACC1 PC

SC

KD

**C** Liver

$p = 7.408\text{E-}7$

$p = 0.0484$

Signal Intensity [a.u.]

CT Mut CT Mut
(n=12) (n=12) (n=12) (n=12)

ACC1 PC

**D** HepG2 Cells

$p = 1.931\text{E-}7$

$p = 0.0165$

Signal Intensity [a.u.]

SC KD SC KD
(n=16) (n=13) (n=16) (n=17)

ACC1 PC

**E**

 SC KD

ACC1

α-Tub

**F**

$p = 0.0072$

ACC1 Expression [a.u.]

SC KD
(n=3) (n=3)

**G** Beta Oxidation

$p = 0.0333$

Liver CPT1 Activity [nmol/min/g]

WT Mut/Mut
(n=3) (n=3)

**H**

KIF12 Deficiency → ACC1 Excess → Excessive Maronyl-CoA

Decrease in Beta Oxidation

Increase in Lipid Production

Liver Steatosis

?

**Figure 5.  KIF12 deficiency resulted in upregulation of ACC1 and PC.**

(A–D) ACC1 and PC immunofluorescence microscopy of *Kif12^{mut/mut}* mouse liver cryosections (A) and HepG2 cells transduced with scrambled control (SC) and KIF12-knockdown (KD) vectors (B), with respective statistics (C and D). Scale bars, 100 μm in A and 20 μm in (B). Error bars, mean ± SEM. Welch's *t* test. Biological replicates, 12 optical fields from 3 mice (C); and individual cells (D). (E, F) ACC1 immunoblotting of HepG2 cell lysates transduced with SC and KIF12-KD vectors (E) with statistics (F). Error bars, mean ± SEM. Welch's *t* test. Biological replicates, individual dishes. (G) Comparison of CPT1 activity of wild type (WT) and *Kif12^{mut/mut}* mouse livers, as an indicator of mitochondrial beta oxidation of fatty acids. Error bars, mean ± SEM. Unpaired single-sided Welch's *t* test. Biological replicates, mouse individuals. (H) The working hypothesis. KIF12 deficiency may affect beta oxidation of the liver probably through the elevation of the ACC1 product, Malonyl-CoA, as well as upregulating lipid production, to promote steatosis. Corresponding to Fig. EV5.

methods of CRISPR-Cas9 technology (Tanaka et al, 2023) as depicted in Fig. 2A–C. We altered *c*.1084_1089 from ACCCGG to ACGTGA to introduce *p*.T362T silent mutation and *p*.R363X nonsense mutation simultaneously, to provide premature termination of the protein translation as well as a *Pml1* restriction site for easy restriction fragment length polymorphism (RFLP) genotyping. To achieve a double-strand break, the CRISPR target 5′-CTTGGGACCCTGTG GCCGGG-3′ (Fig. 2B) was inserted into the Cas9-expressing *pX330* vector (Addgene plasmid #42230). A homologous recombination template, ssODN (single-strand oligodeoxynucleotide) was synthesized (Fig. 2C). 10 μg/mL of ssODN and 5 μg/mL of *pX330-Kif12* vector were simultaneously injected into the pronuclei of one-cell stage C57BL/6J mouse embryos, which were transferred into pseudopregnant ICR mouse oviducts. Four out of 96 pups (male #7; females #52, #82, #89) were identified to be positive for homologous recombination without random integrations according to genomic PCR/RFLP and next-generation sequencing that was 100% match to the expected sequence (Fig. EV2B). They were maintained in specific pathogen-free environment under a 14/10-h light/dark cycle and were freely provided with water and standard mouse chow (CE-2, CLEA Japan); in accordance with the institutional guidelines and approval of University of Tsukuba and The University of Tokyo. They were maintained by intercrossing under a C57BL/6 genetic background (CLEA Japan).

## Mouse histology and blood biochemistry

For liver histology, 12-week-old mice were sacrificed by cervical dislocation or deep anesthesia, and their livers were fixed with 10% buffered formalin or with 4% paraformaldehyde (PFA) as previously described (Wisse et al, 2010); and the sera were collected from the pericardial space. The livers were processed for paraffin sectioning at 5–10 μm thickness, stained with hematoxylin and eosin (H&E) and/or Masson's trichrome at Septosapie Co., Ltd., Tokyo, Japan, and observed with a Leica DM290HD microscope and a Keyence BZ-700 microscope at IRCN, The University of Tokyo, according to standard methods (Bancroft and Stonard, 2019). The blood biochemistry was performed by the standard methods (Oriental Yeast Co., Ltd., Japan).

## Antibodies and labeling reagents

Anti-mouse KIF12-PRD polyclonal antibodies against a middle portion (#88; RRID:AB_2892103, 1:200) (Yang et al, 2014) and a C-term portion of the PRD domain (#E28700-C; RRID: AB_3665302, 1:200) were generated in rabbits at AbClonal/Yurogen and used for immunoblotting. A goat anti-KIF12 antibody was obtained from Santa Cruz Biotechnology (#SC-48558, RRID:AB_2131108) and used for immunofluorescence microscopy at a 1:200 dilution. A mouse monoclonal anti-hepatocyte specific antigen antibody (#SC-58693, RRID:AB_781327, 1:1000) and a

mouse monoclonal anti-COP1 antibody (#SC-166799, RRID:AB_2178894, 1:1000) were also obtained from Santa Cruz Biotechnology. A rabbit anti-COP1 antibody was obtained from St. John's Laboratory (#STJ92423, RRID:AB_3665303, 1:200). Rabbit anti-HA-tag (#561, RRID:AB_591839), rabbit anti-GFP (#598, RRID:AB_591816, 1:1000), and mouse monoclonal anti-multi-ubiquitin (#D071-3, RRID:AB_592938, 1:1000) antibodies were obtained from MBL; a rabbit anti-ACC1 antibody (#21923-1-AP, RRID:AB_11042445, 1:1000) was obtained from Proteintech; a rabbit anti-PC antibody (#NBP1-49536SS, RRID:AB_10011588, 1:1000 for immunoblotting and 1:300 for immunocytochemistry and PLA) was obtained from Novus Biologicals; a mouse monoclonal anti-GAPDH antibody (#016-25523, RRID:AB_2814991, 1:1000) was obtained from FUJIFILM-Wako; a mouse monoclonal anti-α-tubulin antibody DM1A (#F2168, RRID:AB_476967, 1:500) was obtained from Sigma-Aldrich; a mouse monoclonal anti-GFP antibody (#A11120, RRID:AB_221568, 1:50), Alexa-labeled secondary antibodies (1:500) and Alexa-555-phalloidin (#A34055, 1:1000) were obtained from ThermoFisher; HRP-labeled protein A (#NV9120V, 1:5000) and an HRP-labeled anti-mouse IgG antibody (#NA9310V, RRID: AB_772193, 1:5000) were obtained from GE Healthcare; and an alkaline phosphatase-labeled anti-rabbit IgG antibody (#59298, 1:1000) was obtained from Cappel. MitoTracker Red CMXRos (#M7512) and LysoTracker Red DND-99 (#L7528) reagents were obtained from Thermo Fisher and used at 1:10,000 dilution.

## HepG2 cells

HepG2 cells were obtained from Riken Cell Bank and cultured following the provider's protocol. The cells were maintained in D-MEM (#044-29765, FUJIFILM-Wako) supplemented with 10% fetal bovine serum (FBS; #F7524, Sigma-Aldrich) and penicillin-streptomycin (#15140-122, Thermo Fisher, 1:100) at 37 °C in a 5% $CO_2$ atmosphere. The cells were passaged twice a week at a 1:6 ratio on poly-L-lysine-coated 10 cm plastic dishes (#150466, Thermo Fisher) with a half-diluted 0.5% trypsin-EDTA solution (#15400-054, Thermo Fisher) with Hank's balanced salt solution (HBSS; #084-08345, FUJIFILM-Wako) for 5 min at 37 °C.

To generate a lipid-loading model for KIF12-PRD-rescue, HepG2 cells were treated by 1 μM U0126 (#WDM0235, FUJI-FILM-Wako) overnight and then incubated for 3.5 h with 360 μM oleic acid (#151-03425, FUJIFILM-Wako) as previously described (Tsai et al, 2007). Then, the cells were transduced with either mCit or mCit-KIF12-PRD adenoviral expression vectors for 48 h and subjected to LipidTOX staining.

For investigating the overnutrition-based KIF12 expression regulation, HepG2 cells were loaded with 200 μM oleic acid plus bovine serum albumin (BSA) as previously described (Ohtsubo et al, 2011; Pappas et al, 2002). They were incubated for 24 h and served for the experiments.

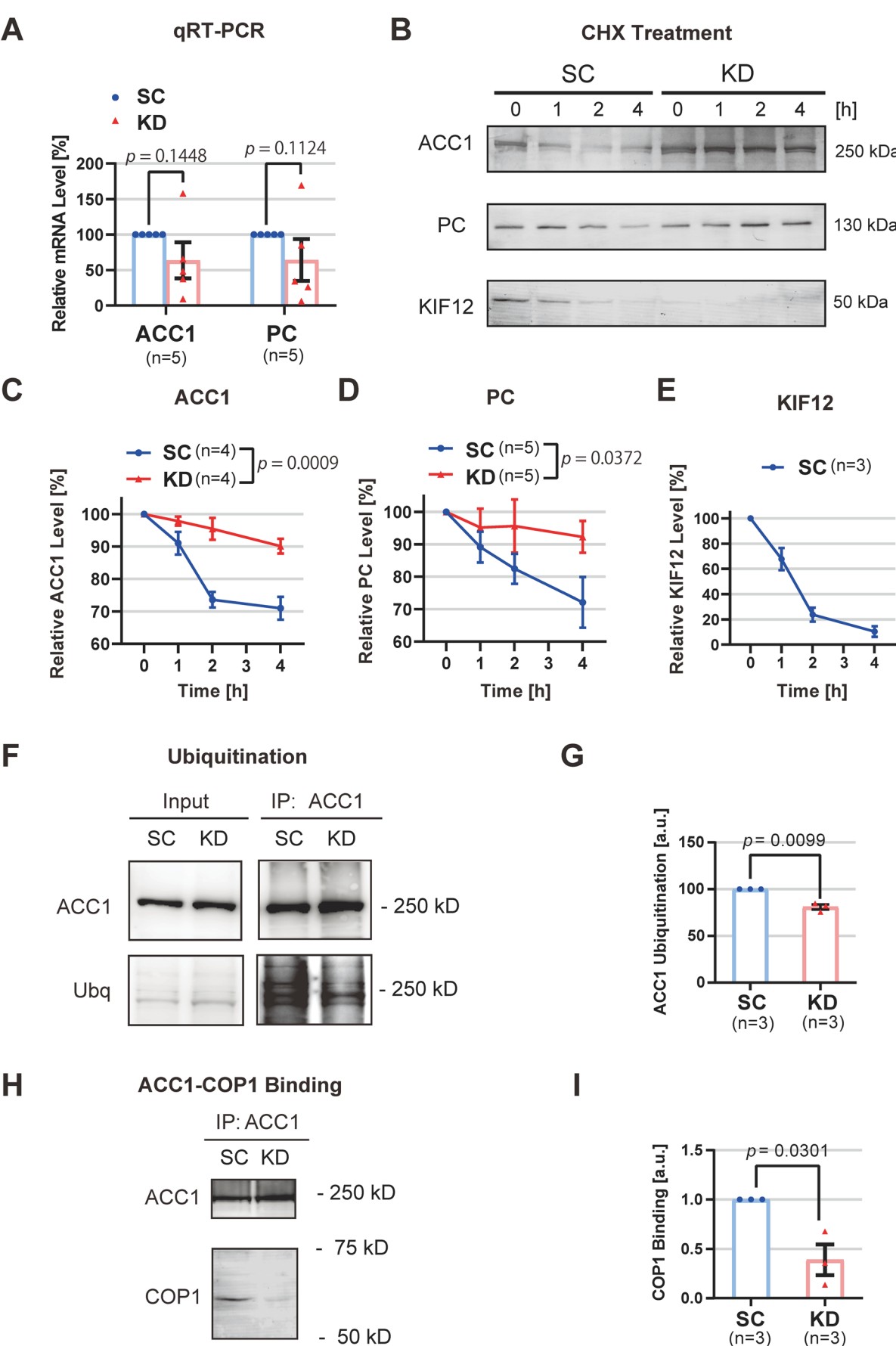

◀ **Figure 6.  KIF12 deficiency reduces turnover of ACC1 and PC.**

(A) qRT-PCR for *ACC1* mRNA in HepG2 cells transduced with KD and SC vectors; normalized by the GAPDH levels. Error bars, mean ± SEM. One-tailed Welch's paired *t* test. Biological replicates, individual wells. Corresponds to Table EV1. (B–E) Protein turnover assays. ACC1, PC, and KIF12 immunoblotting of SC- and KD-transduced HepG2 cell lysates after treatment by 100 μg/mL cycloheximide (CHX) for the indicated periods (B); accompanied by the respective statistics (C–E). Error bars, mean ± SEM. Two-way ANOVA. Biological replicates, individual dishes. (F, G) ACC1 ubiquitination assay of a KIF12-KD system in HepG2 cells in the presence of proteasomal and lysosomal inhibitors, according to ACC1 immunoprecipitation (F) and its statistics (G). Error bars, mean ± SEM. Paired one-sided Welch's *t*-test. Biological replicates, individual dishes. Note that the ACC1 level among the KIF12-KD system became similar by the inhibitor treatment but ACC1-associated ubiquitin level was higher in KD cell lysates. (H, I) Co-immunoprecipitation between ACC1 and COP1 among the KIF12-KD system treated by proteasomal and lysosomal inhibitors for 8 h (H) and its statistics (I). Error bars, mean ± SEM. Paired one-sided Welch's *t*-test. Biological replicates, individual dishes. Corresponding to Fig. EV6.

For BioID proximity labeling assays, HepG2 cells were stably transfected with bait plasmids using Lipofectamine LTX reagent (#2250332, Thermo Fisher) under the selection of 200 μg/mL G418 (#10131035, Thermo Fisher) for 2 weeks. The expression and enrichment of the transfected cells were verified by immunofluorescence microscopy using an anti-HA-tagged antibody (MBL).

## Transdifferentiation

Human fibroblasts were obtained by skin biopsy under written consent and maintained by passaging twice a week with human fibroblast medium (D-MEM/F12 [#11330-032, Thermo Fisher] supplemented with 10% FBS, 4 ng/mL bFGF [#233-FB-010/CF, R&D Systems], 100 μM 2-mercaptoethanol [#137-0682, FUJIFILM-Wako], 1× MEM nonessential amino acids [#11140, Thermo Fisher], and 1:100 penicillin/streptomycin [#15140, Thermo Fisher]).

For lentiviral cocktail preparation, 293FT cells (Thermo Fisher) were transfected with *pWPI.1-FOXA3, pWPI.1-HNF4A, pWPI.1-HNF1A*, and *pWPI.1-Large T* vectors (kindly provided by Dr. Lijian Hui, Shanghai Institute for Biological Sciences); *psPAX2 (*packaging plasmid; #12260, Addgene); and *pMD2.G* (envelope-expressing plasmid; #12259, Addgene) at a 5:3:2 ratio using Lipofectamine 3000 reagent (#15338-100, Thermo Fisher) following the manufacturers' protocols. The culture medium was collected at 24 h and 48 h after transfection and cleared by centrifugation at 2000 rpm for 10 min at room temperature (RT; #LC-122, TOMY) to prepare the lentiviral cocktail.

Transdifferentiation was basically conducted as described previously (Huang et al, 2014). Human fibroblasts ($2 \times 10^5$) were seeded in a collagen-coated 6-well tissue-culture dish (#140675, Thermo Fisher), transduced with 10% lentiviral cocktail for 24 h, incubated with human fibroblast medium for 72 h and then with hepatocyte maintenance medium (D-MEM/F12 supplemented with bovine serum albumin [BSA; 2 g/L], ZnCl$_2$ [3.9 μM], ZnSO$_4$·7H$_2$O [2.6 μM], CuSO$_4$·5H$_2$O [0.8 μM], MnSO$_4$ [0.16 μM], galactose [2 g/L], ornithine [0.1 g/L; #150-003211, FUJIFILM-Wako], proline [0.03 g/L; #161-04602, FUJIFILM-Wako], nicotinamide [0.61 g/L; #141-01202, FUJIFILM-Wako], insulin-transferrin-selenium [ITS-G; 1×; #090-06741, FUJIFILM-Wako], transforming growth factor α [TGFα; #AF-100-16A, PeproTech, 40 ng/mL], epidermal growth factor [EGF; #AF-100-15, PeproTech, 40 ng/mL], dexamethasone [10 μM; #047-18863, FUJIFILM-Wako], and 1% FBS) for 15–18 days. The hepatocyte-like identity was verified according to assessment of morphological transformation and the expression of hepatocyte-specific antigen by immunofluorescence microscopy.

## Pharmacology

TOFA (#10005263, Cayman Chemical) was dissolved in DMSO at 2 mg/mL as a stock solution. HepG2 cells were transduced with

miRNA vectors for 48 h, treated by TOFA at 2 μg/mL for 24 h and fixed for neutral lipid staining.

To examine the KIF12 expression levels upon lipid loading, $2 \times 10^4$ HepG2 cells were plated on 804G-matrix-coated 35 mm glass-bottomed dishes (#D11530H; Matsunami, Japan) as previously described (Yang et al, 2014). They were then loaded for 24–48 h with low-fat-BSA-alone (A-7030, Sigma-Aldrich), or with BSA-conjugated oleic acid (#151-03425, FUJIFILM-Wako) at 200–500 μM that had been prepared as described (Ohtsubo et al, 2011; Pappas et al, 2002). Then, they were fixed with 2% PFA/0.1% glutaraldehyde (GA)/PBS for immunofluorescence microscopy with *z*-projection imaging at a 0.5 μm step size using a 40×/1.4 plan-apochromat oil-immersion lens on a spinning-disc confocal microscope (Yokogawa/ZEISS), to be quantified with the ImageJ software (Schneider et al, 2012), as described in detail in the following sections. For qRT-PCR, the same loading procedure was executed on a 96-well format.

## Neutral lipid quantification

For staining of neutral lipids, the cells or cryosections were fixed with 2% PFA/0.1% GA in PBS for 10 min at 37 °C, rinsed twice with PBS, incubated with BODIPY 493/503 (#D3922, Thermo Fisher, 1:1000) in PBS (Nakamuta et al, 2005) in PBS or HCS LipidTox (#H34477, Thermo Fisher, 1:1000) in PBS (Liu et al, 2009) for 30–60 min at RT, and then washed 3 times with PBS (in the case of BODIPY staining). They were mounted and observed with a semi-superresolution confocal laser scanning microscope (LSM780-Airyscan, ZEISS). They were quantified using ImageJ and GraphPad PRISM 8–10 software.

## Organelle labeling

HepG2 cells were transduced with mCit-KIF12-PRD for 2 days. For labeling mitochondria and lysosomes, the cells were live-stained by 1:10,000-diluted MitoTracker Red CMXRos and LysoTracker Red DND-99, respectively, in the culture medium, incubated for 30 min, and fixed with 0.1% glutaraldehyde/2% paraformaldehyde/PBS at 37 °C for 10 min, rinsed with PBS, and observed using a LSM780-Airyscan laser scanning confocal microscope (ZEISS).

## Expression vectors

The cloning of mouse *Kif12* cDNA and construction of tagRFP-labeled *Kif12* (#KD5) or scrambled control (SC) RNA interference (RNAi) adenoviral vectors and a mCitrine-tagged RNAi-immune expression vector *mCitrine-KIF12im* have been described previously (Yang et al, 2014).

Construction of the adenoviral expression vectors for the KIF12 truncation series was performed as follows by Dr. Wenxing Yang. KIF12

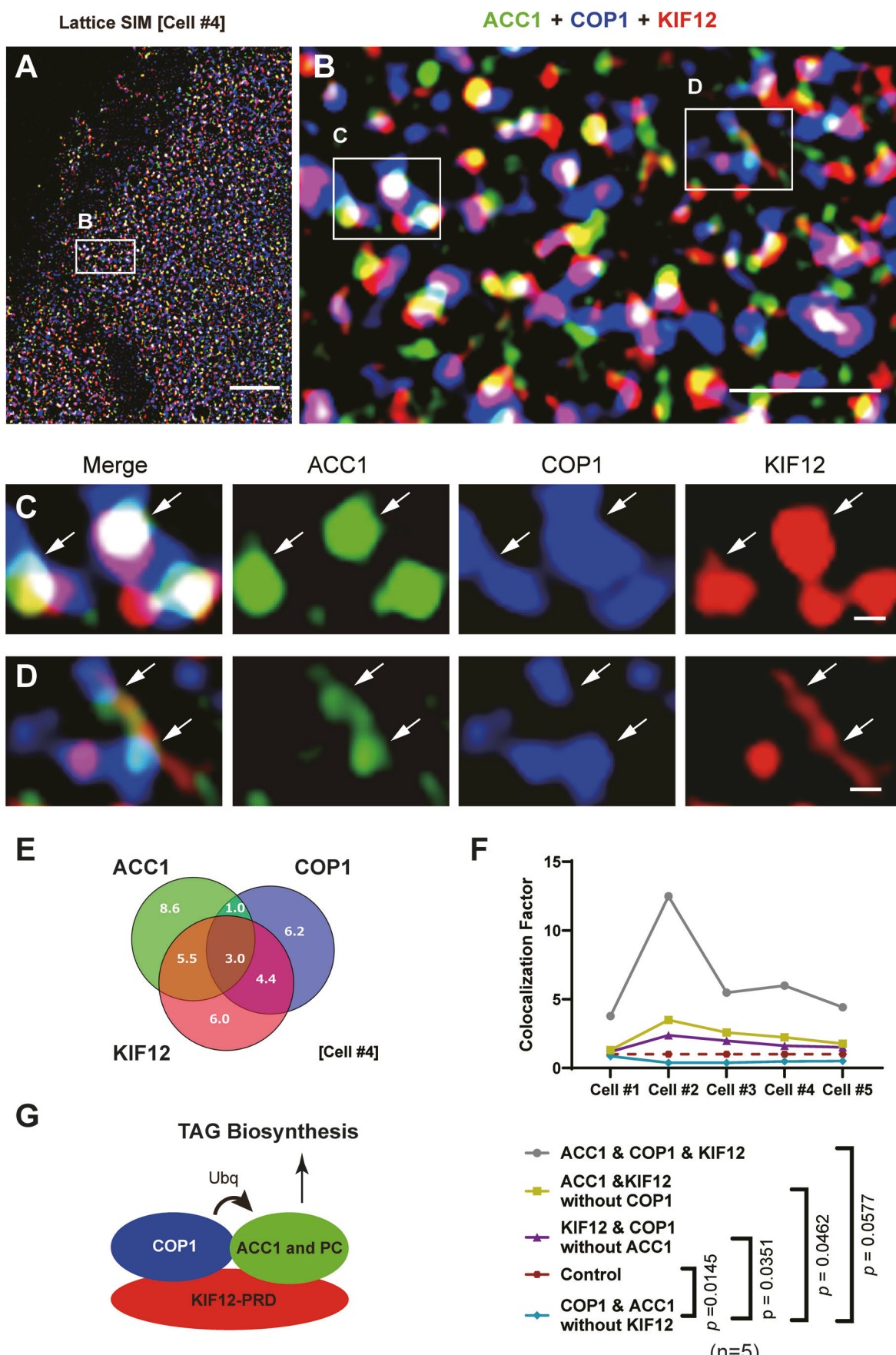

◄ **Figure 7. Ternary association of KIF12, ACC1, and COP1 proteins on microdroplets.**

(A–D) Triple immunofluorescence superresolution microscopy of a HepG2 cell (Cell #4) against ACC1 (green), COP1 (blue), and KIF12 (red), using lattice SIM at the respective magnifications. Arrows, triple-labeled spots. Reproduced in 5 cells. Scale bars, 1 μm (A and B) and 100 nm (C and D). (E) Venn diagram of the percentage of each color area (Cell #4). Note that the percentages of KIF12–ACC1 colocalization without COP1 and KIF12–COP1 colocalization without ACC1 are higher than that of COP1–ACC1 colocalization without KIF12, suggesting that KIF12 intermediates between COP1 and ACC1. (F) Colocalization analysis among 5 cells (Cells #1–5). Colocalization factor (CF) was calculated as the ratio of actual colocalization area over expected colocalization area, which was deduced by multiplying the percentages of each color area. CFs larger than 1 (Control) represent significant colocalization. One-way ANOVA against COP1 & ACC1 without KIF12. Biological replicates, individual cells. (G) The working hypothesis. KIF12 intermediates between the neutral lipid synthesizing enzymes and COP1, and facilitates the ubiquitination and turnover of the enzymes to negatively regulate the liver lipidosis.

fragments from the head, stalk, PRD, and tail were amplified by PCR using the enzyme KOD-Plus-Neo (#KOD-401, TOYOBO) from the *mCitrine-KIF12im* vector with the *K12 Head F* and *K12 Head R; K12 Stalk F* and *K12 Stalk R; K12 PRD F* and *K12 PRD R;* and *K12 Tail F* and *K12 Tail R* primer pairs (Table EV1), respectively. They were digested with the *XhoI* and *SalI* restriction enzymes (except the head fragment, which was digested with *HindIII* and *SalI*; New England Biotech), ligated with the *pmCitrine-C1* vector (Yang et al, 2014), subjected to verification of the nucleotide sequence and the expression of fluorescence, and transferred to a ViraPower Adenoviral Expression System (Thermo Fisher) according to the manufacturer's protocol. The expression of recombinant proteins of the expected lengths was verified by immunoblotting using an anti-GFP antibody. To generate the *pPRD-tagRFP* and *pPRD-GST* expression vectors, *KIF12-PRD* cDNA was recovered from the *pmCit-PRD* vector with the *BglII* and *SalI* enzymes and ligated with the *ptagRFP-C* vector (#FP141, Evrogen) and *pGEX-4T-1* (#28-9545-49, Cytiva) vector, respectively, using these restriction sites.

For generating the *pPRD-BioID2* expression vector, cDNA encoding the KIF12-PRD was amplified with the *N-BAM-ATG* and *C-BAM-NoStop* primers, and ligated with the *MCS-BioID2-HA* vector (kindly provided by Dr. Kyle Roux through Addgene, #74224) (Kim et al, 2016) at the *BamHI* restriction site, and the directions of the inserts were verified by Sanger sequencing.

Human *ACC1* cDNA (Svensson et al, 2016) was a generous gift from Dr. Ruben Shaw (Salk Institute, USA). To generate *EGFP*-tagged ACC1 expression vectors, the *ACC1* cDNA was amplified by PCR using the *ACC-GFP-F/ACC-GFP-R* and *GFP-ACC-F/GFP-ACC-R* primer sets. The cDNA was then ligated with *pEGFP-N1* and *pEGFP-C1* vectors (Clontech) with *SacI* and *SmaI* sites, to generate *pACC-EGFP* and *pEGFP-ACC* expression vectors, respectively.

Human *PC* cDNA (RefSeq: BC011617) was purchased from Sino Biological Inc. To generate *EGFP*-tagged PC expression vectors, the *PC* cDNA was amplified by PCR using the *PC-GFP-F* and *PC-GFP-R;* and *GFP-PC-F* and *GFP-PC-R* primer sets. The cDNA was then ligated with *pEGFP-N1* and *pEGFP-C1* vectors with *EcoRI* and *BamHI* sites, to generate *pPC-EGFP* and *pEGFP-PC* expression vectors, respectively. The constructs were verified by Sanger sequencing before administration to cells.

shRNA knockdown plasmid vectors for *ACC1* (*ACACA*) and *COP1* genes were constructed by Vector Builder. shRNA sequence 5′-GAATCCTCATTGGCCTATAATCTCGAGATTATAGGCCAA TGAGGATTC-3′ for *ACC1* gene and 5′-GCTAACAGTCAGGGTA CAATTCTCGAGAATTGTACCCTGACTGTTAGC-3′ for *COP1* gene were expressed by human U6 small nuclear 1 promoter. It was then ligated with human PGK promoter-driven TagBFP2:-T2A:Puro selection marker, and flanked by piggybac 5′- and 3′-ITR sequences (Li et al, 2005).

## Transfection and transduction

For in vitro adenoviral transduction, HepG2 cells were plated on poly-L-lysine-coated glass-bottomed 35-mm dishes (#D11130H, Matsunami). They were transduced with scrambled RNAi, knockdown RNAi, and/or expression vectors using CsCl-purified adenoviral vectors for 48 h.

For shRNA studies, HepG2 cells were plated on 35-mm glass bottom dishes (Matsunami) at $1–3 \times 10^4$ cells/cm², co-transfected with the respective shRNA plasmid and a hyPBase expression vector plasmid (Vector Builder) on DIV1, transduced with KIF12-KD/SC/PRD adenovirus on DIV2, and fixed to process for immunofluorescence microscopy on DIV6.

## qRT–PCR

For qRT–PCR, HepG2 cells were plated in a 96-well dish and transduced with RNAi vectors for 48 h or treated by oleic acid for 24 h. RNA extraction and cDNA synthesis were performed with a SuperPrep Cell Lysis and RT Kit for qPCR (#SCQ-101, TOYOBO) according to the manufacturer's protocol. cDNA from 1 μg of total RNA was subjected to real-time PCR with the *ACC1-qpcr-F* and *ACC1-qpcr-R; PC-qpcr-F* and *PC-qpcr-R; KIF12-qpcr-F* and *KIF12-qpcr-R; GAPDH-qpcr-F* and *GAPDH-qpcr-R* primers (Table EV1), and SYBR Premix Ex Taq™ (#RR420A, TaKaRa) on a LightCycler 480 system (Roche) at the WINGS-LST collaborative laboratory (The University of Tokyo Graduate School of Medicine). The program consisted of 10 s of preincubation at 95 °C followed by 45 cycles of amplification at 96 °C for 10 s, 60 °C for 15 s and 72 °C for 20 s. The data were normalized against *GAPDH* amplification as previously described (Tanaka et al, 2016) and cases with scores 2 SD above/below the mean were excluded from statistical analyses.

## Fluorescence microscopy

Immunofluorescence microscopy was carried out as previously described (Homma et al, 2003; Yang et al, 2014) using Can Get Signal Immunostain Immunoreaction Enhancer Solution A (#NKB-501, TOYOBO) following the manufacturer's protocol. ACC1-EGFP-transfected cells were fixed with 2% PFA/0.1% GA/PBS at 37 °C for 10 min. The stained cells were subjected to observation with a semi-superresolution laser scanning confocal microscope (LSM780-Airyscan, ZEISS) equipped with a ZEISS 40×/1.4 plan-apochromat oil-immersion lens or a ZEISS 100×/1.46 a-plan-apochromat oil-immersion lens; a spinning-disc confocal microscope (ZEISS-Yokogawa, model CSU-W1 on an Axio Observer.Z1 microscope) equipped with a ZEISS 40×/1.4 plan-apochromat oil-immersion lens or a ZEISS 100×/1.46 a-plan-apochromat oil-immersion lens and a Teledyne Photometrics Prime 95B sCMOS camera); a laser scanning confocal microscope (TCS SP8

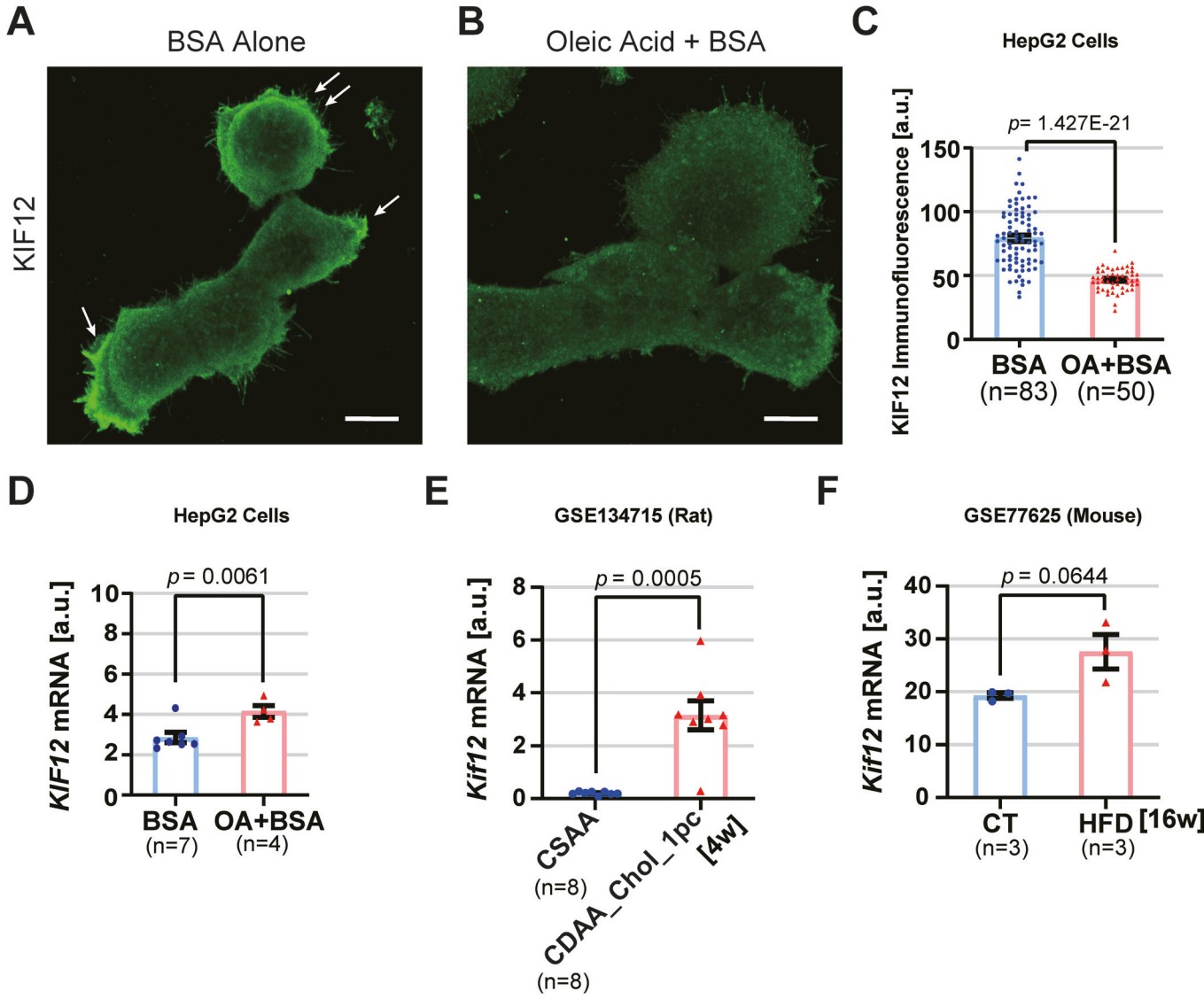

**Figure 8.   Fatty acid loading downregulates KIF12 protein expression.**

(**A–C**) KIF12 immunofluorescence microscopy of HepG2 cells after loading BSA alone (**A**) or 200 µM oleic acid (OA) + BSA for 24 h (**B**); and its quantification (**C**). Note that oleic acid loading led to KIF12 downregulation. Scale bars, 10 µm. Error bars, mean ± SEM. Welch's *t* test. Biological replicates, individual cells. Arrows, peripheral KIF12 accumulations. (**D**) qRT-PCR of KIF12 on the above conditions. Note that *KIF12* mRNA was paradoxically increased by oleic acid treatment, possibly through a negative feedback mechanism. Error bars, mean ± SEM, Biological replicates, individual wells. One-tailed Mann–Whitney test. Repeated twice. (**E, F**) Examples of transcriptome analysis comparing the normalized read numbers of *Kif12* mRNA in in vivo samples from rodent MASH models. (**E**) A choline-deficient and cholesterol-supplemented diet-fed rat model, corresponding to Sample #3 of Table EV2. Error bars, mean ± SEM. Biological replicates, mouse individuals. Welch's *t* test. Accession No. GSE134715. (**F**) A high-fat-diet (HFD)-fed mouse model, corresponding to Sample #14 of Table EV2. Error bars, mean ± SEM. Welch's *t* test. Biological replicates, mouse individuals. Accession No. GSE77625. Corresponds to Fig. EV7.

FALCON, Leica) equipped with a Leica 63×/1.40 HC plan-apochromat oil-immersion lens #11506350 and HyD detectors or an Nikon A1 confocal microscope at IRCN, The University of Tokyo; or a lattice-SIM superresolution microscope (Elyra7, ZEISS, according to the kind direction of Prof. Fumiyoshi Ishidate, iCeMS, Kyoto University) equipped with 642/561/488-nm lasers and a 100×/1.46 a-plan-apochromat oil-immersion lens under the default settings for SIM image processing with ZEISS ZEN software.

For 3D fluorescence imaging of cells, *z*-stack images at 0.22-µm intervals in dual colors were obtained with a spinning disc microscope equipped with a 40×/1.4 lens using MicroManager 1.4 software to calculate the 3D projection views using ImageJ 1.48 v or 1.52i software with the Fiji plugin.

## Colocalization analysis

For the colocalization analysis of superresolution microscopy, we modified a previously described colocalization analysis method (Jaskolski et al, 2005). First, we changed the images into binary using ImageJ/Fiji software and measured the area with each color

and that with overlapping colors to depict them in a Venn diagram. As a null hypothesis, we assumed if every color pixel was randomly localized and pixels with overlapping colors may emerge in a random manner. We calculated the Colocalization Factor (CF) as the ratio of the actual area of overlapping colors over the expected value of the area of overlapping colors, which was calculated from the occupying areas in the region of interest. If the CF was more than 1, colocalization likelihood is higher, so we plotted the CFs for each pair of colors in each cell and compared the tendency of colocalization among the colors by one-way ANOVA. According to this comparison, we can predict which of three proteins is most likely to be serving as a scaffold for the other two proteins.

## Triglyceride assay

For the triglyceride assay, fresh liver tissue was homogenized with 25 mg/mL in ultra-pure water, dissolved with chloroform-methanol (2:1 in volume) solution to 1.25 mg/mL, shook at RT for 30 min, vortexed vigorously and centrifuged at 3000 rpm for 15 min. Then, the upper layer was discarded, and the lower layer was collected and dried by a vacuum drying equipment. Tissue was redissolved in isopropanol, and subjected to a triglyceride assay kit (#632-50991, Wako), following the manufacturer's protocol.

## CPT activity assay

The CPT activity assay was performed basically as described previously (Hao et al, 2021). Approximately 0.1 g of mouse liver was extracted and its CPT activity was measured using a CPT activity assay kit (#BR5001028, Bioleaper, Shanghai, China) according to the manufacturer's protocol.

## Tag proximity ligation assay

For the tag proximity ligation assay, HepG2 cells were transfected with mCitrine-tagged expression vectors for 18 h and then subjected to a proximity ligation assay with a mouse anti-GFP antibody (Thermo Fisher, 1:50) and a rabbit anti-ACC1 antibody (1:100; in Can Get Signal Solution A) using the Duolink system (#DUO92101, Sigma-Aldrich) according to the manufacturer's protocol. The cells were observed with a ZEISS LSM780-Airyscan microscope equipped with a 40×/1.4 oil-immersion lens.

## Immunoblotting

Immunoblotting was performed on both cell lysates and mouse liver lysates, using Can Get Signal Immunoreaction Enhancer Solutions 1 and 2 (TOYOBO), ECL Prime western blotting detection reagents (#GERPN2236, Sigma-Aldrich), and a Quant LAS4000 Mini image analyzer (Cytiva) according to the manufacturers' protocols, as previously described (Yang et al, 2014). Alternatively, we applied alkaline phosphatase-conjugated secondary antibodies (Cappel, 1:1000) to develop color with BCIP and NBT (Sigma-Aldrich) in 50 mM Tris-HCl (pH 8.0), 100 mM NaCl, and 2 mM $MgCl_2$.

## BioID proximity labeling

BioID proximity labeling was performed as previously described (Roux et al, 2013). Briefly, HepG2 cells stably expressing KIF12-PRD-BioID2 or BioID2 proteins were seeded in 10-cm plates and exposed to biotin (Sigma-Aldrich, #B4501; 50 mM) for 18 h. The lysates were affinity-purified with Dynabeads MyOne Streptavidin C1 beads (#65001, Thermo Fisher) overnight and subjected to SDS–PAGE. Specific bands were recovered and subjected to LC/MS analyses at FUJIFILM-Wako Pure Chemicals and verified by immunoblotting. This KIF12-PRD-BioID2 worked better than BioID2-KIF12-PRD or KIF12-PRD-BirA vectors.

## GST pulldown

GST pulldown assays were performed as previously described (Jordens et al, 2001). The *E. coli* strain BL21 was transformed with the expression vectors *pPRD-GST* and *pGEX-4T-1*, respectively, cultured in 20 mL of LB/0.1% ampicillin at 37 °C for 6–8 h and further cultured at 24 °C overnight after adding 20 µl of 0.5 M isopropyl β-D-1-thiogalactopyranoside (IPTG). The bacterial pellet was harvested with 1 mL of BugBuster Protein Extraction Reagent (#70584, Merck) containing protease inhibitors, and the supernatant was loaded onto 20 µL of Glutathione Sepharose 4B beads (#17075601, Cytiva). 0.3 g of mouse liver was triturated against 3 mL of the homogenizing buffer [150 mM NaCl, 0.1 M HEPES pH 7.4, 1% digitonin, protease inhibitors], and cleared by centrifugation thrice at 15,000 rpm for 10 min at 4 °C. The beads were then rinsed once with 500 mM NaCl and thrice with the homogenizing buffer, mixed with 1 mL of the supernatant, rotated at 4 °C for 2 h, washed thrice with the homogenizing buffer, and eluted with 60 µL of 1× Laemmli's sample buffer for SDS–PAGE.

## Immunoprecipitation

IP, tag-IP, and coIP were performed basically as described previously (Yang et al, 2014). HepG2 cells were transduced with the appropriate RNAi and/or expression vectors for 48 h. In the ACC1 ubiquitination assay, the cells were further treated by 200 µg/mL chloroquine (#C6628, Sigma-Aldrich) and 300 ng/mL MG132 (#P1102, Enzo) for 8–10 h to block the proteasomal and lysosomal degradation pathways. The cells were washed once with cold PBS and then triturated 10 times by passaging through a Myjector 29 G insulin syringe (Terumo) with 1 mL of ice-cold lysis buffer (150 mM NaCl, 10 mM HEPES, 5 mM $MgSO_4$, 1 mM EGTA, cOmplete-mini protease inhibitor cocktail [Roche]). The lysate was then cleared by centrifugation twice at $3000 \times g$ for 10 min at 4 °C. One microgram of the primary antibody was mixed with 20 µl of MACS-Protein A microbeads (#5131211268, Miltenyi Biotec) for 30 min on ice and incubated with the supernatant for 2 h at 4 °C with rotation. The lysate was then passed through an µMACS column (#130-042-701, Miltenyi Biotec), and the column was rinsed five times with 100 µL of the buffer at 4 °C. The immunoprecipitants were then eluted with 60 µl of heated 1× Laemmli's sample buffer (5 min, 3 times) and subjected to immunoblotting for detection of the protein of interest.

## Protein turnover assay

A protein turnover assay with cycloheximide (CHX) was performed as previously described (Yang et al, 2014). HepG2 cells were seeded in a 24-well chambered glass-bottomed dish (#C501301R, Eppendorf) and transduced with either SC or KD RNAi adenoviral

vectors. The cells were treated by CHX (100 μg/mL) for the indicated periods to block protein synthesis. Crude cell lysate extracts were subjected to immunoblotting.

## Transcriptome analyses

For assessing the *Kif12* expression changes over the MASH progression, we compared the normalized read number of *Kif12* mRNA in genomics analyses on the Gene Expression Omnibus (GEO) database from National Center for Biotechnology Information (NCBI) website (https://www.ncbi.nlm.nih.gov/geo/) (Clough et al, 2024). They were statistically analyzed using Welch's *t* test otherwise it had not been statistically tested previously. The examined datasets were summarized in Table EV2.

## Statistics

All experiments were reproduced at least twice. The sample size was estimated to appropriately reflect the nature of the phenomenon. The images were quantified using ImageJ software (Schneider et al, 2012) and Prism 8–10 software (GraphPad). All quantitative data are presented as the mean ± SEM. Cases with scores 2 SD above/below the mean were excluded from statistical analyses in qRT-PCR experiments. Statistical analysis was performed using one-way and two-way ANOVA, Welch's *t* test, Mann–Whitney test, as indicated.

## Study approval

The human study was performed under approval from Hadassah Hebrew University Medical Center and The University of Tokyo Graduate School of Medicine (#G10017). All animal experiments were conducted with male mice under the University of Tokyo's Restrictions on Animal Experimentation under approval from the Institutional Animal Care and Use Committee of the University of Tokyo Graduate School of Medicine (#M-P15-118 and #M-P20-092).

# Data availability

The materials and data will be available upon request. The original dataset was deposited to Mendeley Data, V2, https://doi.org/10.17632/k4jrc3hy3f.2.

The source data of this paper are collected in the following database record: biostudies:S-SCDT-10_1038-S44318-025-00366-8.

# Peer review information

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

## Acknowledgements

We thank Lijian Hui (Shanghai Institutes for Biological Sciences) for the *pWPI* vectors carrying FOXA3, HNF1A, HNF4A expression units, Kyle Loux for the *MCS-BioID2-HA* vector, Ruben Shaw (Salk Institute) for the *ACC1* cDNA, Lou Yang from AbClonal/Yurogen for antibody production; Seiya Mizuno and

Satoru Takahashi (Transborder Medical Resource Center, Tsukuba University) for the knockin mouse generation; Yasuyuki Morishita (Univ Tokyo) for histopathology, Fumiyoshi Ishidate and Minako Kengaku (iCeMS, Kyoto Univ) for the lattice SIM microscope, Satoru Kondo and Masumi Asahara (IRCN, Univ Tokyo) for the Leica FALCON microscope, Mayumi Kaneda (FUJIFILM-Wako Pure Chemicals) for mass-spectrometry, Yoshihiro Kita (Univ Tokyo) for TAG measurements, Yukinori Okada, Yotaro Kudo, Hiroyuki Aburatani (Univ Tokyo) for valuable discussions. We also thank Wenxing Yang, Tadayuki Ogawa, Noriko Homma, Shuzo Hasegawa, Yiming Li, Shimpei Goto, Hiromi Sato, Haruyo Fukuda, Nobuhisa Onouchi, Tsuyoshi Akamatsu, and previous members of the Hirokawa Laboratory for their valuable help and discussion. This study was supported by JSPS KAKENHI grants JP23000013 and JP16H06372 (NH), 20K06634 (YT), a research fund supported by JEOL (NH), a grant-in-aid of the Uehara Memorial Foundation (YT), a grant-in-Aid of the ONO Medical Research Foundation (YT), and by a research grant from Japan Agency for Medical Research and Development 24nk0101645h0003 (YT).

## Author contributions

**Asieh Etemad**: Investigation; Methodology; Writing—original draft; Writing—review and editing. **Yosuke Tanaka**: Conceptualization; Supervision; Funding acquisition; Investigation; Methodology; Writing—original draft; Writing—review and editing. **Shuo Wang**: Investigation; Writing—review and editing. **Mordechai Slae**: Investigation; Methodology; Writing—original draft; Writing—review and editing. **Mutaz Sultan**: Investigation; Methodology; Writing—original draft; Writing—review and editing. **Orly Elpeleg**: Supervision; Funding acquisition; Writing—original draft; Writing—review and editing. **Nobutaka Hirokawa**: Conceptualization; Supervision; Funding acquisition; Project administration; Writing—review and editing; and direction.

Source data underlying figure panels in this paper may have individual authorship assigned. Where available, figure panel/source data authorship is listed in the following database record: biostudies:S-SCDT-10_1038-S44318-025-00366-8.

## Disclosure and competing interests statement

The authors declare no competing interests.

# Expanded View Figures

**Figure EV1. Symptoms of human pedigrees with KIF12 mutations.**

(**A**) MRI images of Patient 2 indicating hepatosplenomegaly. Corresponds to Fig. 1D. (**B**) Summary of blood biochemistry of Patients 2 (Blue) and 3 (Red). Glu glucose, CRE creatinine, TP total protein, ALB albumin, γ-GTP gamma-glutamyl transpeptidase, T-BIL total bilirubin, D-BIL direct bilirubin, LDH lactate dehydrogenase, UR_AC uric acid, CHOL cholesterol. Corresponds to Fig. 1F–H. (**C**) Pedigree of Patient 3's family. Corresponds to Fig. 1F–H.

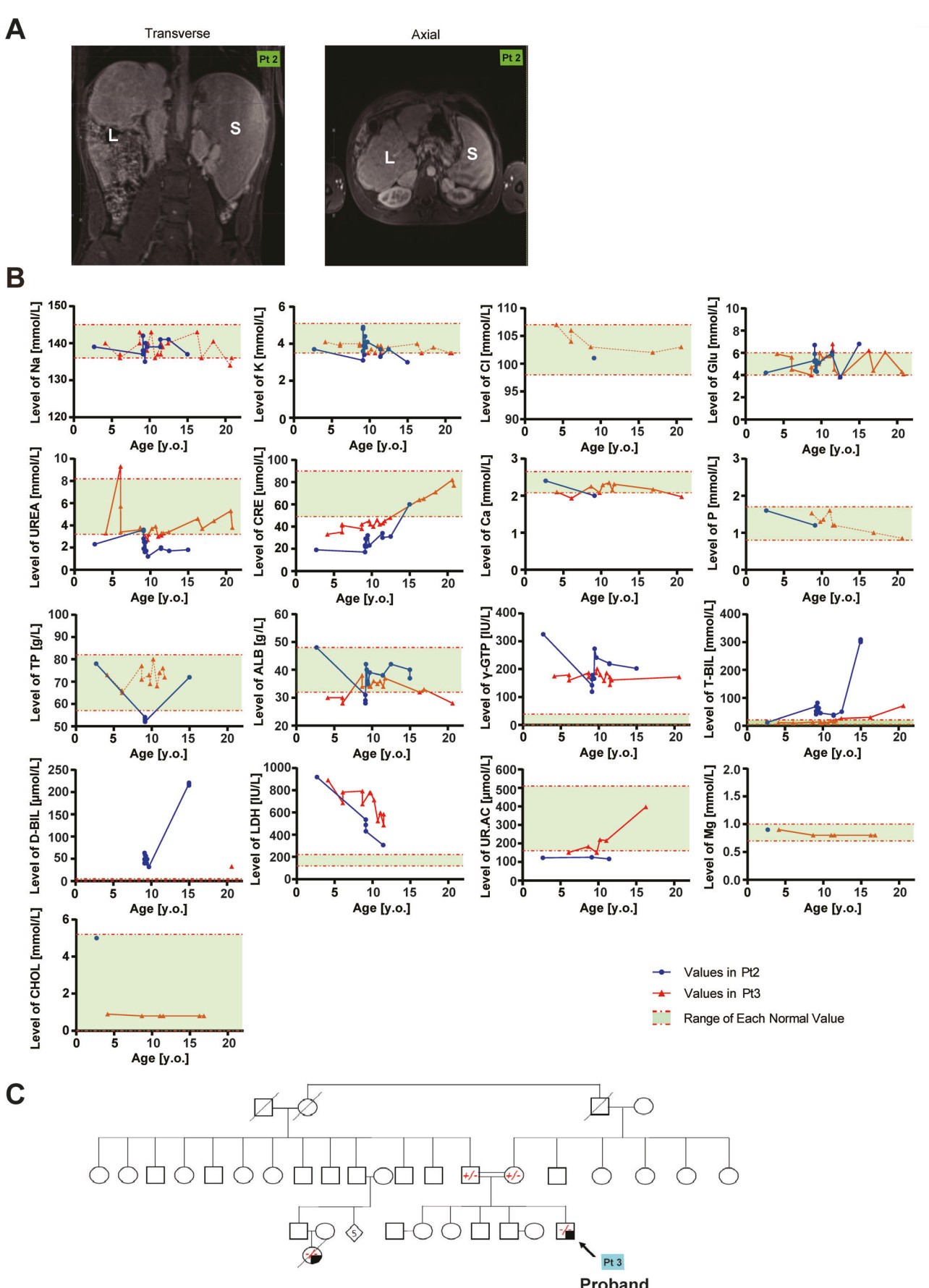

**A**  Amino acid sequence alignments of KIF12 variants and mice

**B**

**C**

RFLP

**D**

**E**

KIF12 IHC

**Figure EV2.  Generation of KIF12 mouse model.**

(**A**) Amino acid sequence alignments of human KIF12 variants and a mouse KIF12 sequence (Yang et al, 2014). Corresponds to Fig. 2A. (**B**) Next-generation sequencing of the F0 pups. KIF12PM07_S39 (#7) was turned out to be the correct one. Corresponds to Fig. 2B,C. (**C**) RFLP analysis by *Pml1* restriction digestion. WT, wild type. M, molecular weight markers. Corresponds to Fig. 2B,C. (**D**, **E**) Immunohistochemistry of F0 *Kif12^mut/mut^* mouse liver indicating a significant decrease in KIF12 immunofluorescence in the liver of a strain #7 mouse (**D**), accompanied by quantification (**E**). Scale bar, 50 µm. Error bars, mean ± SEM. Welch's *t* test. Biological replicates, 9 optical fields from 3 independent F0 individuals. Corresponds to Fig. 2B,C.

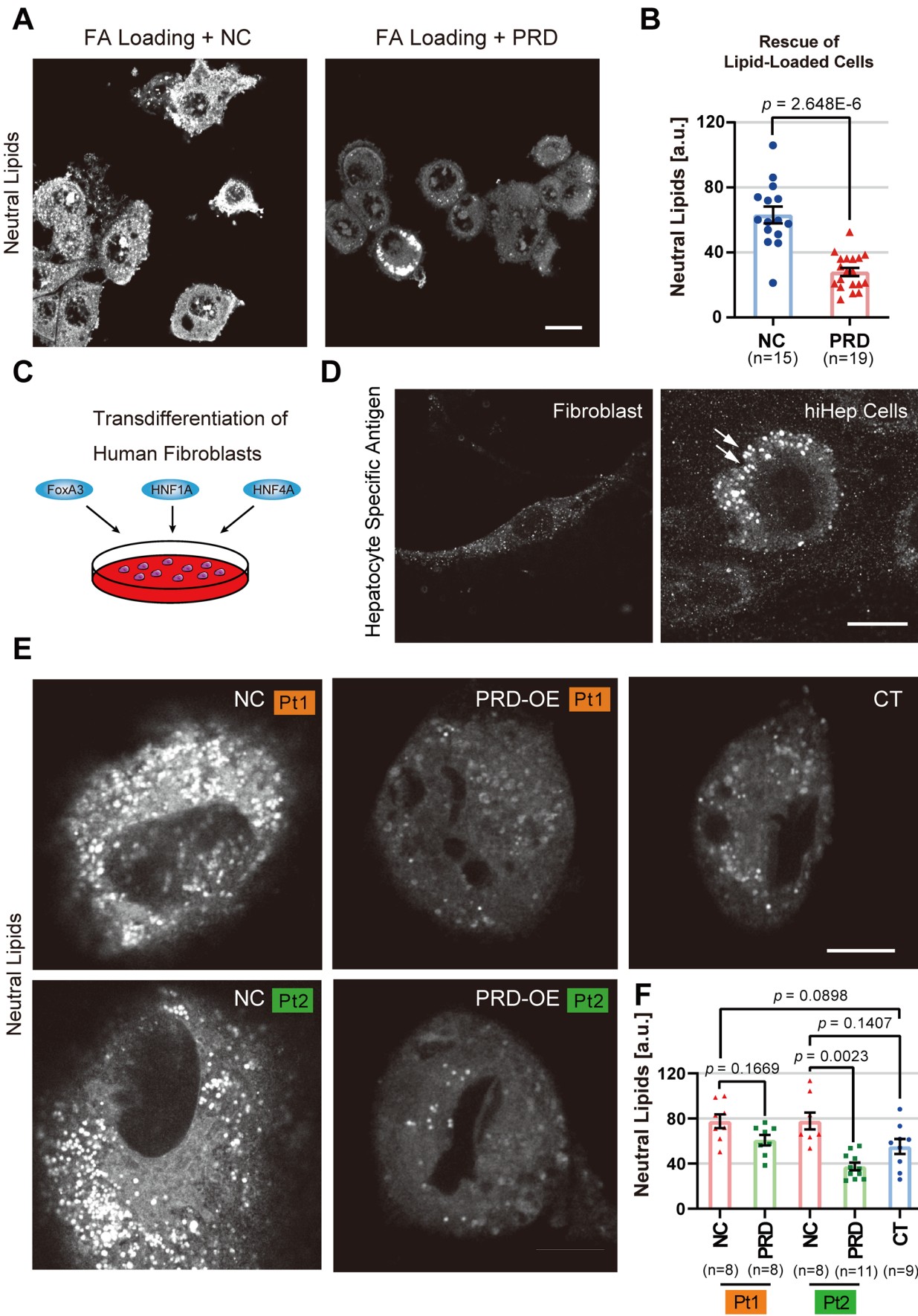

◀ **Figure EV3. KIF12-PRD overexpression ameliorates lipid accumulation in general, corresponding to Fig. 3.**

(A, B) LipidTOX staining of U0126/oleic-acid-treated (FA loading) HepG2 cells transduced with the mCit-alone (negative control; NC) and mCit-PRD (A) and its quantification (B). Error bars, mean ± SEM. Welch's $t$ test. Biological replicates, individual cells. (C, D) hiHep cell transdifferentiation from human fibroblasts, indicated by an experimental scheme (C) and immunofluorescence microscopy against Hepatocyte-Specific Antigen (HSA) immunocytochemistry (D). Scale bar, 10 μm. Arrows in (D), HSA signals. (E, F) LipidTOX neutral lipid staining of patients' (Pt1 and Pt2) and control (CT) hiHep cells without (NC) and with KIF12-PRD overexpression (PRD; E), accompanied by statistics (F). Scale bar, 10 μm. Error bars, mean ± SEM. One-way ANOVA. Biological replicates, individual cells.

NT PRD OE

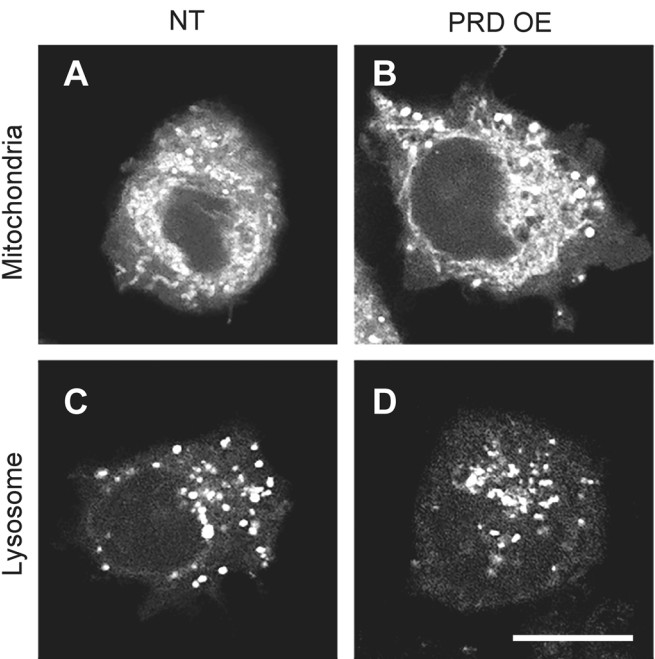

**Figure EV4. The effect of PRD overexpression on mitochondrial morphology, corresponding to Fig. 3.**

(A, B) The morphology of mitochondria (A, B) and lysosomes (C, D) of HepG2 cells; without (A, C) or with (B, D) mCit-PRD overexpression. Note that PRD overexpression tended to increase mitochondrial complexity. Scale bar, 20 µm.

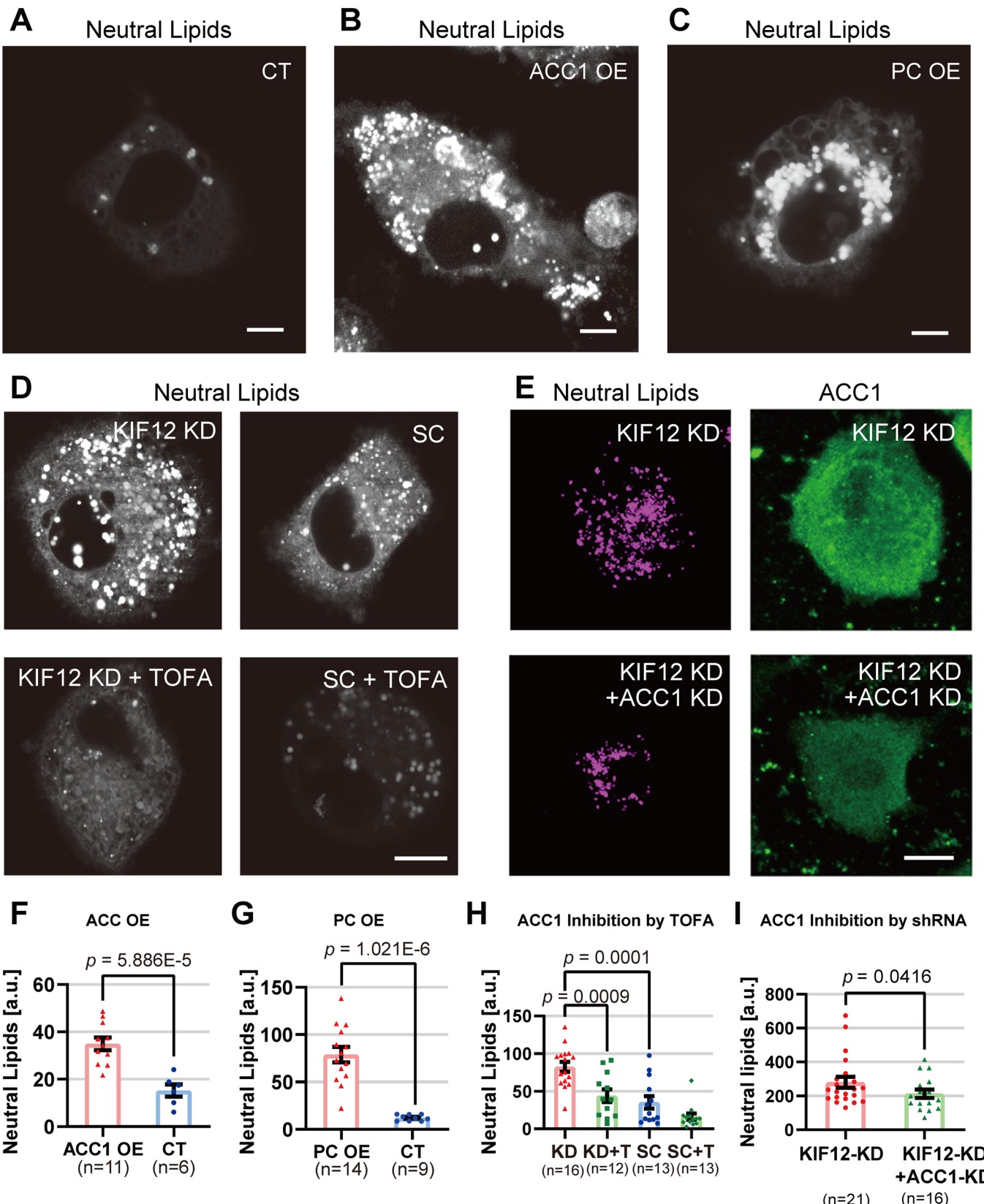

◄ **Figure EV5. The relevance of ACC1 upregulation in lipid accumulation in KIF12-deficient cells, corresponding to Fig. 5.**

(A–C) LipidTOX neutral lipid staining of HepG2 cells overexpressing EGFP alone (CT; **A**), ACC1-EGFP (**B**), and PC-EGFP (**C**). Scale bars, 10 μm. (D, E) LipidTOX neutral lipid staining (**D**, **E**) and ACC1 immunofluorescence (**E**) of HepG2 cells treated by KIF12-KD or SC miRNAs; together with the ACC1 inhibitor, TOFA (2 μg/mL for 24 h; **D**), or with ACC1-KD shRNA (**E**). Scale bars, 20 μm. Repeated twice. (F–I) Statistics of neutral lipid staining levels (**F** and **G** for **A–C**; **H** for **D**; **I** for **E**). Error bars, mean ± SEM. One-sided nonpaired Welch's *t* test (**F**, **G**), one-way ANOVA (**H**), and one-sided Mann–Whitney's test (**I**). Biological replicates, individual cells. T, TOFA in (**H**).

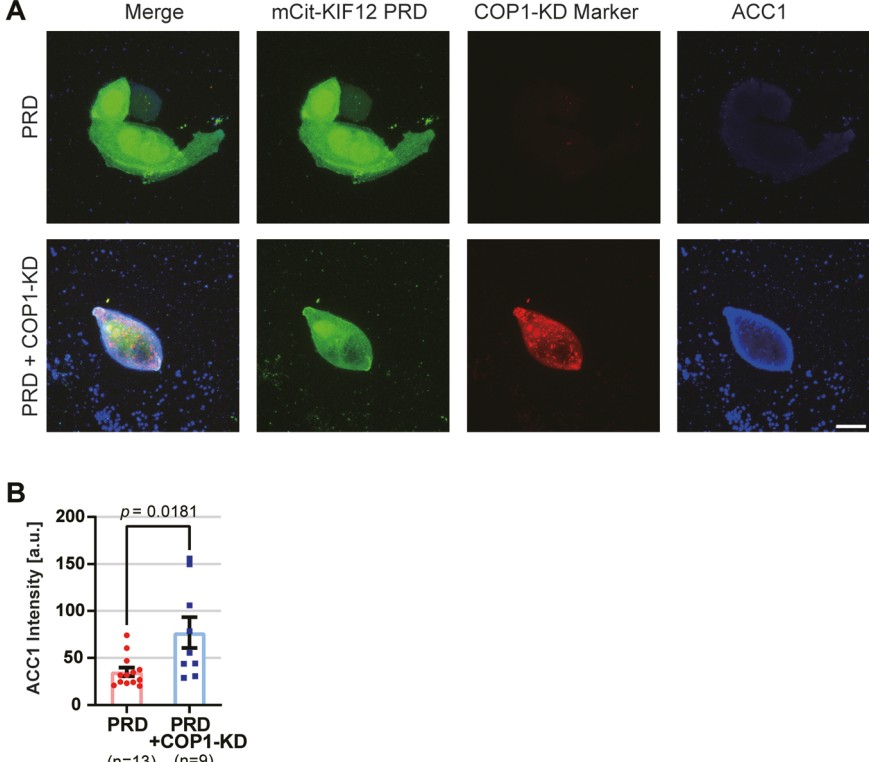

Figure EV6. The relevance of COP1 in ACC1 turnover against lipidosis, corresponding to Fig. 6.

(A) ACC1 immunofluorescence with transfection markers of mCit-PRD-overexpressing HepG2 cells, without (PRD) or with treatment by COP1-KD shRNA (PRD + COP1-KD). Scale bars, 20 µm. Note that COP1 deficiency increased the ACC1 level. (B) Statistics of (A). Error bars, mean ± SEM. One-sided nonpaired Welch's *t* test. Biological replicates, mouse individuals. Corresponds to Fig. 6H,I.

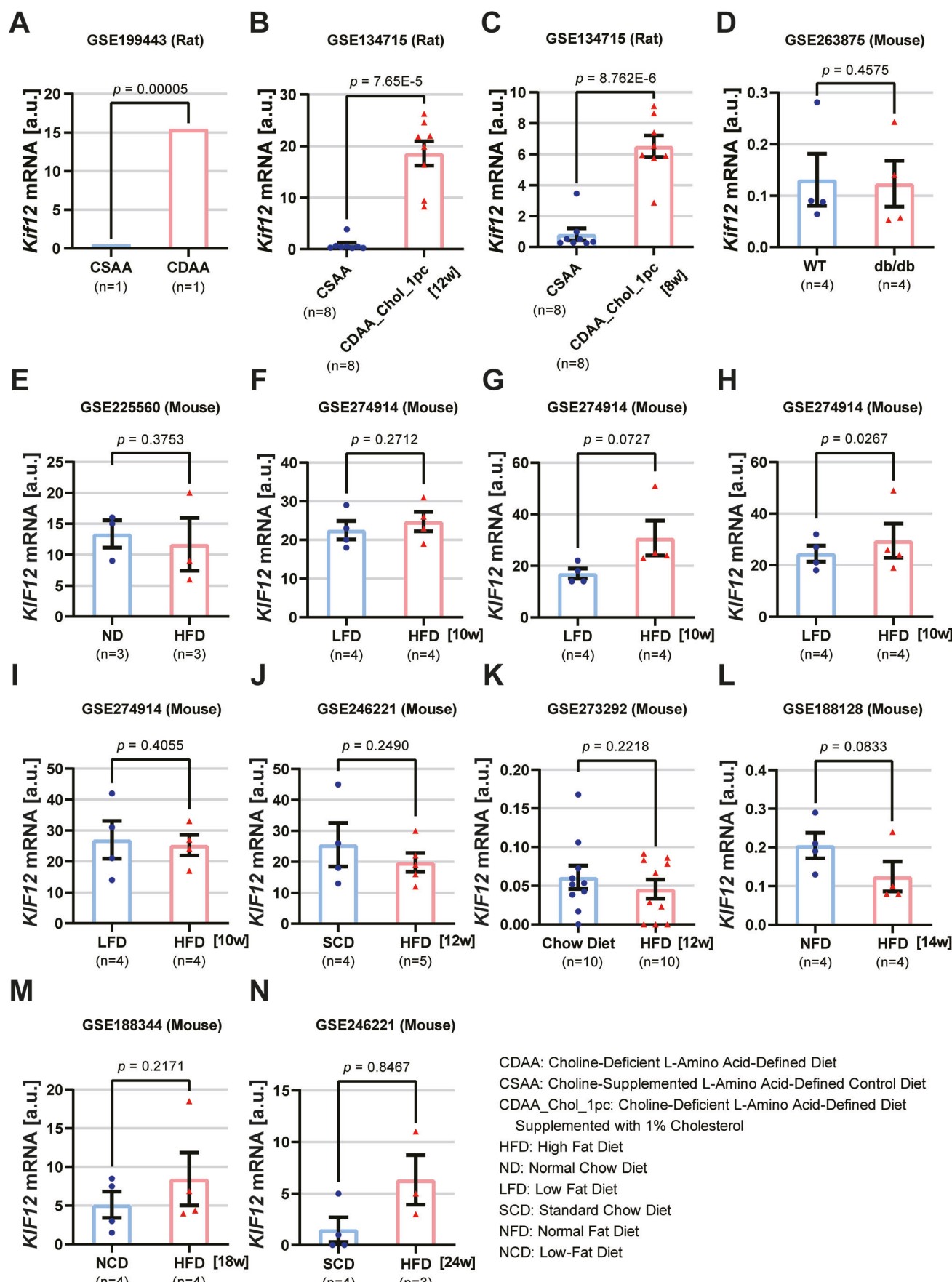

CDAA: Choline-Deficient L-Amino Acid-Defined Diet
CSAA: Choline-Supplemented L-Amino Acid-Defined Control Diet
CDAA_Chol_1pc: Choline-Deficient L-Amino Acid-Defined Diet
    Supplemented with 1% Cholesterol
HFD: High Fat Diet
ND: Normal Chow Diet
LFD: Low Fat Diet
SCD: Standard Chow Diet
NFD: Normal Fat Diet
NCD: Low-Fat Diet

◀ **Figure EV7. Summary of *Kif12* transcriptome analyses on MASH rodent models.**

(**A–N**) Normalized *Kif12* RNA sequence read number comparison of in vivo samples from rodent MASH models (**A–C**, rats fed with choline-deficient diet; **D**, *db/db* mouse; **E–N**, high-fat diet-fed mice for the indicated periods), according to the indicated public database accession numbers. Error bars, mean ± SEM. Welch's *t* test. Corresponds to Table EV2 and Fig. 8E,F.

