## [Peer Review File · The EMBO Journal]

Mutations in the kinesin KIF12 promote MASH in humans and mice by disrupting lipogenic enzyme turnover

Asieh Etemad, Yosuke Tanaka, Shuo Wang, Mordechai Slae, Mutaz Sultan, Orly Elpeleg, and Nobutaka Hirokawa

Corresponding author: Nobutaka Hirokawa (hirokawa@m.u-tokyo.ac.jp)

Review Timeline:

Submission Date:	14th May 24
Editorial Decision:	12th Jun 24
Revision Received:	9th Nov 24
Editorial Decision:	4th Dec 24
Revision Received:	22nd Dec 24
Accepted:	8th Jan 25

Editor: Ieva Gailite

Transaction Report:

Dear Nobutaka,

I would like to thank you for submitting your manuscript for consideration by the EMBO Journal. We have now received a full set of reviewer reports, which are included below for your information.

As you will see from the reports, the reviewers find the study of interest, while also pointing out several experiments and further quantifications that would be required to strengthen the conclusiveness of the study, in particular regarding regulation of COP1-dependent ACC1 ubiquitination by KIF12, the proposed bridging role of KIF12 in the formation of the KIF12-ACC1-COP1 complex, and further evidence for therapeutic efficacy of the KIF12 PRD in an animal model. Based on the interest expressed in the reports, I would like to invite you to address the issues raised by all referees in a revised manuscript. I think it would be useful to discuss the revision in more detail via email or phone/videoconferencing - please let me know which option you prefer.

We generally allow three months as standard revision time. As a matter of policy, competing manuscripts published during this period will not negatively impact on our assessment of the conceptual advance presented by your study. However, please contact me as soon as possible upon publication of any related work to discuss the appropriate course of action. Should you foresee a problem in meeting this three-month deadline, please contact us to arrange an extension.

When preparing your letter of response to the referees' comments, please bear in mind that this will form part of the Review Process File and will therefore be available online to the community. For more details on our Transparent Editorial Process, please visit our website: <https://www.embopress.org/page/journal/14602075/authorguide#transparentprocess>. Please also see the attached instructions for further guidelines on preparation of the revised manuscript.

Please feel free to contact me if you have any further questions regarding the revision. Thank you for the opportunity to consider your work for publication, and I look forward to your revision.

With best wishes,

Ieva

- a point-by-point response to the referees' comments, with a detailed description of the changes made (as a word file).
- a word file of the manuscript text.
- individual production quality figure files (one file per figure)
- a complete author checklist, which you can download from our author guidelines (<https://www.embopress.org/page/journal/14602075/authorguide>).

- Expanded View files (replacing Supplementary Information)

We realize that it is difficult to revise to a specific deadline. In the interest of protecting the conceptual advance provided by the work, we recommend a revision within 3 months (10th Sep 2024). Please discuss the revision progress ahead of this time with the editor if you require more time to complete the revisions.

Referee #1:

The authors have investigated a very unique aspect of the Kif12 kinesin motor, and how it helps in the reduction of lipids in the liver. This article is quite relevant to potential peptide-based new therapies for NASH. PRD domain and tail domain of Kif12 are identified as most relevant in this aspect. There is strong data that Kif12 acts as a scaffolding protein for ACC1 and COP1, which in turn ubiquitinates ACC1 for degradation and thus reduces biosynthesis of lipids to reverse NASH. Particularly nice is the part on patient samples who are genetically predisposed to NASH. Using hiHep cells, HepG2 cells and mouse liver makes a strong impact. Evidence for the above stated ternary complex comes from Tag proximity ligation assay, BioID proximity assay, and Co-IP. Overall, this is a strong contribution. We have two major concern that should be addressed before publication, and several minor concerns listed below

Major concerns

- 1) The difference in liver TAG (Fig 2E) of Mut/Mut mice is barely significant. This is a major point in the study, so this measurement should be repeated with more N to confirm such a difference. TAG was measured with a kit. It should also be measured using other techniques as well (eg TLC, Lipidomics)
- 2) In the section "KIF12 makes a tertiary complex with ACC1 and COP1" authors suggested Liquid-Liquid phase separation to form a non-membranous punctate inside the cell for these three proteins interacting with each other. These results are very weak as they lack of mathematical analysis of the claimed co-localization. Most importantly, authors must show that a non-PRD domain (eg KIF12-Head or KIF12-Stalk) does not show this kind of behavior. Overall, we are quite concerned about this section - we think it should be removed, or otherwise it should be substantiated significantly if it is to be included in this paper.

Minor concerns

- 3) The words "new relevant" in the title are quite unnecessary, they should be removed
- 4) The Authors have not provided related images for control in Fig. 2D. These are mice liver sections, so it should be easy to do so. Comparison with normal liver is needed to appreciate the proposed micro-steatosis, formation of Mallory-Denk bodies, and inflammatory cell infiltration. Control images with H&E staining and Masson trichome staining are recommended.
- 5) The images in Fig. 3A. are very noisy. De-noising the images or reducing the gain voltage, while acquiring the images might help to give clarity to images. Background noise makes it difficult to quantify signals.
- 6) The western blot to show the Kif12 knockdown in Fig. 3D. doesn't seem very reliable since the GAPDH band in both lanes seems to have merged. A sharper band for the western is recommended.
- 7) The Authors claim that Kif12 PRD expression ameliorates fat accumulation by interacting with ACC1 and PC and enhances their degradation rates by ubiquitination. ACC1 and PC are well-known lipid biosynthetic enzymes in hepatocytes. The authors didn't mention how the existing fat which has already been synthesized gets reduced inside the cell. Since biosynthetic enzymes are depleted, biosynthesis of new LDs is blocked but we see that the already-existent LDs also disappear with PRD treatment after fatty acids loading. This may suggest that the mitochondria are activated for β -oxidation with ACC1 depletion due to higher degradation of the protein. An experiment to check for CPT1 activity is recommended in PRD-treated cells.
- 8) In Fig. 6C. the authors have claimed that ACC1 is more localized in the cell periphery in the knockdown condition but in the image the cell looks very distorted from its normal morphology to a round shape. It is not clear if the ACC1 is localized to the periphery or if the cell cytosol is occupying a smaller area due to distortions in cellular morphology resulting in such an effect. The authors need to stain for DAPI/ Hoescht Blue nuclear stain to check for the nucleus-to-cytoplasm ratio.
- 9) In Fig. 6E. there is too much noise in the KD lane for ACC1 staining as given in the western blot image. A clearer band is recommended as noise makes it difficult to quantify.

10) The authors have claimed that despite mRNA expressions being low ACC1 and PC protein levels are high in the KD condition due to post-translational change in the knockdown condition, where the ubiquitination of ACC1 is also low. The authors need to address whether the increase in ACC1 levels inhibits the transcription of new ACC1 mRNA by knocking down as well as over-expressing ACC1 to expect an increase and depletion of mRNA respectively in Fig. 7A.

11) In Fig. 7G. the COP1 protein bands in the western are not clear. COP1 has a molecular weight of approximately 80 kDa but the bands are all seen above 100 kDa even for the original form. A clearer band for COP1 is recommended.

12) The authors have written that Kif12 makes a tertiary complex with ACC1 and COP1 which should be ternary.

13) The authors also need to check for any disruptions in the sub-cellular localization of organelles like mitochondria, lysosomes, ER, and lipid droplets to rule out any adverse effect of PRD on vesicular trafficking by live cell imaging or immuno-staining.

Referee #2:

Summary

In the current study, Eternad et al. explore the role of the kinesin superfamily member KIF12 in the development of Metabolic-Dysfunction Associated Steatohepatitis (MASH). Using whole exome sequencing in three human cases, the authors discovered mutations in KIF12 correlating with markers of liver dysfunction. Mimicking these KIF12 mutations in mice by CRISPR/Cas9 technology led to a MASH-like phenotype in these animals, showing steatosis, formation of Mallory-Denk bodies and fibrosis. Knockdown of KIF12 in HepG2 hepatoma cells also triggered lipid accumulation in these cells, mediated by the proline-rich domain (PRD) within KIF12, which suppressed lipid accumulation upon overexpression in cells. The PRD was found to interact with the lipogenic genes ACC1 and PC. Knockdown of KIF12 induced the upregulation of ACC1 and PC protein expression, mediated by a reduced protein turnover due to diminished ACC1 ubiquitination. Also, KIF12 deficiency diminished the interaction between ACC1 and E3 ubiquitin ligase COP1, while in turn KIF12 was identified as a scaffold protein within a tertiary complex consisting of ACC1-KIF12 and COP1. Overall, the authors conclude that the KIF12-PRD could serve as a new modality in the treatment against MASH by counteracting hepatic lipid accumulation.

General comments

Despite the current approval for resmetirom as a first drug in the treatment of MASH, key mechanistic insights into MASH pathogenesis are still lacking to date and there is still a major unmet clinical need to define further therapeutic modalities. In this respect, the manuscript by Eternad et al. identify an interesting and biomedically relevant pathway with potential therapeutic implications. The strength of this study resides in the use of human material as a starting point of further experiments, documenting high translational relevance. The study employs state-of-the-art technology and the conclusions are generally supported by the experimental data. The manuscript is well-written, concise and well-structured. On the other hand, there are two major concerns that require additional attention by the authors: a) Despite the author's claims, a clear functional evidence for the therapeutic effectiveness of PRD in a relevant animal model is missing. The authors should ideally employ GAN diet-fed animals and then treat these animals with their PRD compound. If peptide stability was insufficient, an AAV-mediated overexpression in hepatocytes could also be sufficient to support their conclusions in the GAN model by then monitoring MASH parameters. b) The authors should provide some data on KIF12 protein expression in different MASH/steatosis mouse models to evaluate the broader implication of their observations. Are KIF12 protein levels elevated in db/db, HFD, GAN, MCD mice and correlated with hepatic lipid levels and ACC1/PC levels? The addition of these data will significantly strengthen the case for publication.

Specific comments

- Introduction: Please do not use the old NASH nomenclature but refer to MASH
- Simple reduction in steatosis will not be sufficient to counteract MASH but also inflammatory reactions have to be diminished. Pls discuss briefly in the discussion section.
- Fig. 6: Please confirm the Tofa data by using additionally ACC1 siRNA.
- Please use COP1 siRNA plus PRD overexpression to show the effect on ACC1 ubiquitination and protein levels.

Referee #3:

This is an interesting study that performs a functional analysis of specific mutations in the kinesin 12 protein found in patients with NASH. The study uses mouse and cultured cell models to show a strong correlation between the mutations and the steatotic phenotype. The functional interactions of KIF12 and two key enzymes in the lipid synthetic pathways are described and convincing. Several suggestions are as follows.

Central Suggestions

The authors try to provide mechanistic insights into how the KIF12 mutations lead to steatosis via providing a scaffolding function to facilitate the ubiquitination of ACC1 by COP1. This, in my view, is the weakest part of the story. While the reduction of the ACC1 and PC in the KDs are convincing (Fig 7C-E) the blots in Fig 7F,G are rather marginal? Can some quantitation be provided and also perhaps some other approaches to test this concept and strengthen this central conclusion? If it's not

ubiquitination then that is fine but the case for or against needs to be stronger than it is as presented

As a continuation to this segment Fig 8 shows some nice high resolution molecular imaging of ACC1, COP1 AND KIF12. This finding could be expanded and strengthened with statistical analysis that quantitates the green, red, yellow and white overlapping areas from multiple areas in multiple cells and from several experiments.

Minor points

- Some of the higher mag histological images could be better such as Fig 1E and 2D.
- The histological images in 6A are much more dramatic than the graph depicts?

Point-by-point response to the reviewer's comments:

Etemad et al.

November 9, 2024

Referee #1:

The authors have investigated a very unique aspect of the Kif12 kinesin motor, and how it helps in the reduction of lipids in the liver. This article is quite relevant to potential peptide-based new therapies for NASH. PRD domain and tail domain of Kif12 are identified as most relevant in this aspect. There is strong data that Kif12 acts as a scaffolding protein for ACC1 and COP1, which in turn ubiquitinates ACC1 for degradation and thus reduces biosynthesis of lipids to reverse NASH. Particularly nice is the part on patient samples who are genetically predisposed to NASH. Using hiHep cells, HepG2 cells and mouse liver makes a strong impact. Evidence for the above stated ternary complex comes from Tag proximity ligation assay, BioID proximity assay, and Co-IP. Overall, this is a strong contribution. We have two major concern that should be addressed before publication, and several minor concerns listed below

Thank you so much for supporting the conceptual advance and physiological relevance of this article. We have done our best to improve this manuscript along with this reviewer's insightful suggestions.

Major concerns

1) The difference in liver TAG (Fig 2E) of Mut/Mut mice is barely significant. This is a major point in the study, so this measurement should be repeated with more N to confirm such a difference. TAG was measured with a kit. It should also be measured using other techniques as well (eg TLC, Lipidomics)

We conducted total n = 6–7 of TAG measurement experiment, which greatly improved

the statistical significance of this study. We could not conduct other suggested studies because of the limited resources, while we will surely continue this direction of research in the future.

Figure 2E. Liver TAG levels. One-sided Welch's *t* test, n = 6–7.

2) In the section "KIF12 makes a tertiary complex with ACC1 and COP1" authors suggested Liquid-Liquid phase separation to form a non-membranous punctate inside the cell for these three proteins interacting with each other. These results are very weak as they lack of mathematical analysis of the claimed co-localization. Most importantly, authors must show that a non-PRD domain (eg KIF12-Head or KIF12-Stalk) does not show this kind of behavior. Overall, we are quite concerned about this section - we think it should be removed, or otherwise it should be substantiated significantly if it is to be included in this paper.

Because this is triple immunofluorescence study, we cannot regrettably perform the suggested truncation experiments. However, we have fully performed detailed mathematical colocalization studies using 5 independent cells. First, we measured the area of each color and wrote a Venn diagram (**Figure 7E**), suggesting that KIF12 is significantly more colocalizing to COP1 and ACC1. Then we multiplied among their ratios to calculate the expected values assuming if the relationship among each color is totally independent. We defined "Colocalization Factor" (CF) as the ratio of actual area over expected value and calculated them for 5 cells. As a result, CF of triple colocalization was extremely high, and those of KIF12-ACC1 or KIF12-COP1 followed (**Figure 7F**). These results provide evidence that those triple colocalization is not a by-chance product,

and also that KIF12 protein acts as a scaffold that connects ACC1 and COP1 (**Figure 7G**). Thank you very much for this insightful suggestion.

Figure 7. (E) Venn diagram of the percentage of each color area (Cell #4). Note that the percentages of KIF12–ACC1 colocalization without COP1 and KIF12–COP1 colocalization without ACC1 are higher than that of COP1–ACC1 colocalization without KIF12, suggesting that KIF12 intermediates between COP1 and ACC1.

(F) Colocalization analysis among 5 cells (Cells #1–5). Colocalization factor (CF) was calculated as the ratio of actual colocalization area over expected colocalization area, which was deduced by multiplying the percentages of each color area. CFs larger than 1 (Control) represent significant colocalization. One-way ANOVA against COP1 & ACC1 without KIF12, n = 5.

(G) The working hypothesis. KIF12 intermediates between the neutral lipid synthesizing enzymes and COP1, and facilitates the ubiquitination and turnover of the enzymes to negatively regulate the liver lipodosis.

Minor concerns

3) The words "new relevant" in the title are quite unnecessary, they should be

removed

Yes, we removed them as suggested. Now the title reads “An anti-MASH pathway in human and mouse livers involving KIF12 kinesin.”

4) The Authors have not provided related images for control in Fig. 2D. These are mice liver sections, so it should be easy to do so. Comparison with normal liver is needed to appreciate the proposed micro-steatosis, formation of Mallory-Denk bodies, and inflammatory cell infiltration. Control images with H&E staining and Masson trichome staining are recommended.

We added H&E and Masson images of wild type mouse livers.

5) The images in Fig. 3A. are very noisy. De-noising the images or reducing the gain voltage, while acquiring the images might help to give clarity to images. Background noise makes it difficult to quantify signals.

6) The western blot to show the Kif12 knockdown in Fig. 3D. doesn't seem very reliable since the GAPDH band in both lanes seems to have merged. A sharper band for the western is recommended.

We improved anti-KIF12 immunofluorescence microscopy and western blotting of SC/KIF12-KD-transduced HepG2 cells.

7) The Authors claim that Kif12 PRD expression ameliorates fat accumulation by interacting with ACC1 and PC and enhances their degradation rates by ubiquitination. ACC1 and PC are well-known lipid biosynthetic enzymes in hepatocytes. The authors didn't mention how the existing fat which has already been synthesized gets reduced inside the cell. Since biosynthetic enzymes are depleted, biosynthesis of new LDs is blocked but we see that the already-existent LDs also disappear with PRD treatment after fatty acids loading. This may suggest that the

mitochondria are activated for β -oxidation with ACC1 depletion due to higher degradation of the protein. An experiment to check for CPT1 activity is recommended in PRD-treated cells.

Thank you for this insightful suggestion. Because the PRD transduction efficiency was not sufficient to make 100% transduction, we performed CPT1 activity assay in KIF12 mutant mouse liver. The CPT1 activity of the mutant mouse was significantly lower than that of wild type mouse, suggesting that KIF12 helps beta oxidation of fatty acids, just as this reviewer insightfully predicted. Thank you very much for this kind suggestion. We added these data as **Figure 5G** and wrote a working hypothesis scheme in **Figure 5H**.

Figure 5. (G) Comparison of CPT1 activity of wild type (WT) and *Kif12^{mut/mut}* mouse livers, as an indicator of beta oxidation of fatty acids. Error bars, mean \pm SEM. * $p < 0.05$, Unpaired single-sided Welch's t test. $n = 3$.

(H) Working hypothesis. KIF12 deficiency may affect beta oxidation of the liver probably through the elevation of the ACC1 product, Malonyl-CoA, as well as upregulating lipid production, to promote steatosis.

8) In Fig. 6C. the authors have claimed that ACC1 is more localized in the cell periphery in the knockdown condition but in the image the cell looks very distorted from its normal morphology to a round shape. It is not clear if the ACC1 is localized to the periphery or if the cell cytosol is occupying a smaller area due to distortions in cellular morphology resulting in such an effect. The authors need to stain for DAPI/ Hoescht Blue nuclear stain to check for the nucleus-to-cytoplasm ratio.

We have improved the ACC1 immunostaining in **Figure 5B** and toned down the description regarding the subcellular localization.

9) In Fig. 6E. there is too much noise in the KD lane for ACC1 staining as given in the western blot image. A clearer band is recommended as noise makes it difficult to quantify.

We improved the anti-ACC1 immunoblotting of SC/KIF12-KD-transduced HepG2 cells.

Figure 5. (E) ACC1 immunoblotting of HepG2 cell lysates transduced with SC and KIF12-KD vectors.

10) The authors have claimed that despite mRNA expressions being low ACC1 and PC protein levels are high in the KD condition due to post-translational change in the knockdown condition, where the ubiquitination of ACC1 is also low. The authors need to address whether the increase in ACC1 levels inhibits the transcription of new ACC1 mRNA by knocking down as well as over-expressing ACC1 to expect an increase and depletion of mRNA respectively in Fig. 7A.

Thank you for this insightful suggestion, but it will be very difficult to interpret the results of any changes in ACC1 transcription in ACC1-knockdown cells. Alternatively, we found a nice previous work suggesting that the enzymatic product acyl-CoA can negatively regulate ACC1 transcription (Faergeman & Knudsen, 1997). This can better explain why the KIF12-KD cells with steatosis should reduce the ACC1 transcription through this negative regulation mechanism. We added this argument in the main text (p. 12).

11) In Fig. 7G. the COP1 protein bands in the western are not clear. COP1 has a molecular weight of approximately 80 kDa but the bands are all seen above 100 kDa even for the original form. A clearer band for COP1 is recommended.

We improved the COP1 western blotting as follows.

The molecular weight of this protein has a high variety between tissues, and the following results are consistent with our results.

12) The authors have written that Kif12 makes a tertiary complex with ACC1 and COP1 which should be ternary.

Thank you for this kind suggestion on the typo. We fixed this.

13) The authors also need to check for any disruptions in the sub-cellular localization of organelles like mitochondria, lysosomes, ER, and lipid droplets to rule out any adverse effect of PRD on vesicular trafficking by live cell imaging or immunostaining.

We performed mitochondria and lysosome staining of PRD overexpressing cells as the following. Interestingly the mitochondria looked expanded throughout the cytoplasm as CPT1-mediated beta oxidation in the mitochondria is expected to increase. We have explicitly described this in the text. Thank you very much for this insightful suggestion.

Figure EV4. The effect of PRD overexpression on mitochondrial morphology, corresponding to Fig. 4

(A and B) The morphology of mitochondria (A and B) and lysosomes (C and D) of HepG2 cells; without (A and C) or with (B and D) mCit-PRD overexpression. Note that PRD overexpression tended to increase mitochondrial complexity. Scale bar, 20 μ m.

Referee #2:

Summary

In the current study, Eternad et al. explore the role of the kinesin superfamily member KIF12 in the development of Metabolic-Dysfunction Associated Steatohepatitis (MASH). Using whole exome sequencing in three human cases, the authors discovered mutations in KIF12 correlating with markers of liver dysfunction. Mimicking these KIF12 mutations in mice by CRISPR/Cas9 technology led to a MASH-like phenotype in these animals, showing steatosis, formation of Mallory-Denk bodies and fibrosis. Knockdown of KIF12 in HepG2 hepatoma cells also triggered lipid accumulation in these cells, mediated by the proline-rich domain (PRD) within KIF12, which suppressed lipid accumulation upon overexpression in cells. The PRD was found to interact with the lipogenic genes ACC1 and PC. Knockdown of KIF12 induced the upregulation of ACC1 and PC protein expression, mediated by a reduced protein turnover due to diminished ACC1 ubiquitination. Also, KIF12 deficiency diminished the interaction between ACC1 and E3 ubiquitin ligase COP1, while in turn KIF12 was identified as a scaffold protein within a tertiary complex consisting of ACC1-KIF12 and COP1. Overall, the authors conclude that the KIF12-PRD could serve as a new modality in the treatment against MASH by counteracting hepatic lipid accumulation.

Thank you very much for this nice summary, but we have not argued that KIF12 can treat against MASH by counteracting hepatic lipid accumulation. The in vivo therapeutic part is far beyond the scope of this paper which we have not explicitly argued in the text of this manuscript at all. We genetically showed functional evidence that KIF12-loss results in MASH through those in vitro data and functionally disrupting human/mouse data. Plus, we argued that KIF12-PRD is a scaffold for ACC1 ubiquitination as a new relevant molecular mechanism. These two points are the main argument of this paper and we have not argued that KIF12 has a therapeutic value onto MASH patients or MASH mouse models.

General comments

Despite the current approval for resmetirom as a first drug in the treatment of MASH, key mechanistic insights into MASH pathogenesis are still lacking to date and there is still a major unmet clinical need to define further therapeutic modalities. In this respect, the manuscript by Eternad et al. identify an interesting and biomedically relevant pathway with potential therapeutic implications. The strength of this study resides in the use of human material as a starting point of further experiments, documenting high translational relevance. The study employs state-of-the-art technology and the conclusions are generally supported by the experimental data. The manuscript is well-written, concise and well-structured.

Thank you so much for this enthusiastic support.

On the other hand, there are two major concerns that require additional attention by the authors: a) Despite the author's claims, a clear functional evidence for the therapeutic effectiveness of PRD in a relevant animal model is missing. The authors should ideally employ GAN diet-fed animals and then treat these animals with their PRD compound. If peptide stability was insufficient, an AAV-mediated overexpression in hepatocytes could also be sufficient to support their conclusions in the GAN model by then monitoring MASH parameters.

Although we deeply understand what this reviewer suggested, but we would ask you for kind understanding and consideration that a preclinical study of KIF12 into a MASH mouse model is far beyond the scope of this paper under the limited time and available resources, as the first author has already left our laboratory to the US. Firstly, establishing effective peptides or AAV constructs in vitro will require repeated trials and errors for several months in prior to the actual in vivo study. In vivo application study will also require several months to reach statistically significant results. If it is fortunately turned out to be effective, we will also need to characterize the downstream pathways by multi-omics studies which will take several months, followed by its cell biological characterization that will take several months. Thus, establishing scientifically meaningful preclinical PoC needs another effort of years, like other examples in this field

(Bian *et al*, 2022; Goedeke *et al*, 2018; Harrison *et al*, 2024; Kim *et al*, 2018; Zhang *et al*, 2021). Accordingly, we have surely toned down the discussion on the clinical relevance of KIF12 (p. 19), as kindly suggested in this insightful comment.

b) The authors should provide some data on KIF12 protein expression in different MASH/steatosis mouse models to evaluate the broader implication of their observations. Are KIF12 protein levels elevated in db/db, HFD, GAN, MCD mice and correlated with hepatic lipid levels and ACC1/PC levels? The addition of these data will significantly strengthen the case for publication.

Thank you for this insightful suggestion. We have compared the KIF12 expression of disease models partly using available public transcriptome data such as in db/db mouse liver (Goyal *et al*, 2017). We conduct analyses on other mouse data as well.

In addition, we have noticed that overnutrition (oleic acid loading) can reduce KIF12 expression in HepG2 cells at a posttranscriptional level. These data will provide basic supporting evidence for future therapeutic approaches.

(E and F) Examples of transcriptome analysis comparing the normalized read numbers of *Kif12* mRNA on in vivo samples from rodent MASH models according to public database records. (E) A choline-deficient and cholesterol-supplemented diet-fed rat model, corresponding to Sample #3 of **Table EV2**. Error bars, mean \pm SEM. n = 8. Welch's *t* test. Accession No. GSE134715. (F) A high-fat-diet (HFD)-fed mouse model, corresponding to Sample #14 of **Table EV2**. Error bars, mean \pm SEM. n = 3. Welch's *t* test. Accession No. GSE77625. Error bars, mean \pm SEM. Corresponds to **Fig. EV6**.

Specific comments

- Introduction: Please do not use the old NASH nomenclature but refer to MASH

Certainly. We have changed NASH into MASH throughout this paper.

- Simple reduction in steatosis will not be sufficient to counteract MASH but also inflammatory reactions have to be diminished. Pls discuss briefly in the discussion section.

We included a discussion that control of inflammatory reactions is an important issue in MASH treatment in general in p. 19 as kindly suggested.

- Fig. 6: Please confirm the Tofa data by using additionally ACC1 siRNA.

We performed ACC1 gene silencing and showed that this silencing reversed lipid droplet accumulation in KIF12-KD HepG2 cells.

- Please use COP1 siRNA plus PRD overexpression to show the effect on ACC1 ubiquitination and protein levels.

We conducted COP1 gene silencing and showed that this silencing reduced PRD-OE-mediated ACC1 turnover.

Referee #3:

This is an interesting study that performs a functional analysis of specific mutations in the kinesin 12 protein found in patients with NASH. The study uses mouse and cultured cell models to show a strong correlation between the mutations and the steatotic phenotype. The functional interactions of KIF12 and two key enzymes in the lipid synthetic pathways are described and convincing. Several suggestions are as follows.

Thank you very much for this supportive comment. We have done our best to improve the manuscript along with this reviewer's insightful suggestions.

Central Suggestions

The authors try to provide mechanistic insights into how the KIF12 mutations lead to steatosis via providing a scaffolding function to facilitate the ubiquitination of ACC1 by COP1. This, in my view, is the weakest part of the story. While the reduction of the ACC1 and PC in the KDs are convincing (Fig 7C-E) the blots in Fig 7F,G are rather marginal? Can some quantitation be provided and also perhaps some other approaches to test this concept and strengthen this central conclusion? If it's not ubiquitination then that is fine but the case for or against needs to be stronger than it is as presented

Thank you so much for this insightful suggestion. We improved the statistical reproducibility of those studies and added supporting evidence for those arguments.

Regarding new **Fig. 6F and G**, we repeated this co-IP experiment with SC/KIF12-KD transduced and inhibitor treated cells more than three times, and statistically compared the levels of ACC1 ubiquitination.

Regarding new Fig. 7H and I, we repeated this experiment using a new COP1 antibody and statistically compared the effect of KIF12 deficiency against COP1-ACC1 interaction.

As a continuation to this segment Fig 8 shows some nice high resolution molecular imaging of ACC1, COP1 AND KIF12. This finding could be expanded and strengthened with statistical analysis that quantitates the green, red, yellow and white overlapping areas from multiple areas in multiple cells and from several experiments.

Thank you very much. Yes. We performed statistical quantification of those images from multiple areas in multiple cells and from several experiments.

Figure 7. (E) Venn diagram of the percentage of each color area (Cell #4). Note that the percentages of KIF12–ACC1 colocalization without COP1 and KIF12–COP1 colocalization without ACC1 are higher than that of COP1–ACC1 colocalization without KIF12, suggesting that KIF12 intermediates between COP1 and ACC1.

(F) Colocalization analysis among 5 cells (Cells #1–5). Colocalization factor (CF) was calculated as the ratio of actual colocalization area over expected colocalization area, which was deduced by multiplying the percentages of each color area. CFs larger than 1 (Control) represent significant colocalization. One-way ANOVA against COP1 & ACC1 without KIF12, $n = 5$.

(G) The working hypothesis. KIF12 intermediates between the neutral lipid synthesizing enzymes and COP1, and facilitates the ubiquitination and turnover of the enzymes to negatively regulate the liver lipodosis.

Minor points

- Some of the higher mag histological images could be better such as Fig 1E and 2D.

Yes, we improved high mag histological images in Fig 2D, and removed it from Fig 1E as the information could be seen the high mag larger image on the right.

• The histological images in 6A are much more dramatic than the graph depicts?

Yes, we substituted some of the images so as to represent the mean values of the quantifications.

References:

- Bian H, Liu YM, Chen ZN (2022) New avenues for NASH therapy by targeting ACC. *Cell Metab* 34: 191-193
- Faergeman NJ, Knudsen J (1997) Role of long-chain fatty acyl-CoA esters in the regulation of metabolism and in cell signalling. *Biochem J* 323 (Pt 1): 1-12
- Goedeke L, Bates J, Vatner DF, Perry RJ, Wang T, Ramirez R, Li L, Ellis MW, Zhang D, Wong KE *et al* (2018) Acetyl-CoA Carboxylase Inhibition Reverses NAFLD and Hepatic Insulin Resistance but Promotes Hypertriglyceridemia in Rodents. *Hepatology* 68: 2197-2211
- Goyal N, Sivadas A, Shamsudheen KV, Jayarajan R, Verma A, Sivasubbu S, Scaria V, Datta M (2017) RNA sequencing of db/db mice liver identifies lncRNA H19 as a key regulator of gluconeogenesis and hepatic glucose output. *Sci Rep* 7: 8312
- Harrison SA, Bedossa P, Guy CD, Schattenberg JM, Loomba R, Taub R, Labriola D, Moussa SE, Neff GW, Rinella ME *et al* (2024) A Phase 3, Randomized, Controlled Trial of Resmetirom in NASH with Liver Fibrosis. *The New England journal of medicine* 390: 497-509
- Kim HY, Kumar H, Jo MJ, Kim J, Yoon JK, Lee JR, Kang M, Choo YW, Song SY, Kwon SP *et al* (2018) Therapeutic Efficacy-Potentiated and Diseased Organ-Targeting Nanovesicles Derived from Mesenchymal Stem Cells for Spinal Cord Injury Treatment. *Nano letters* 18: 4965-4975
- Zhang XJ, Ji YX, Cheng X, Cheng Y, Yang H, Wang J, Zhao LP, Huang YP, Sun D, Xiang H *et al* (2021) A small molecule targeting ALOX12-ACC1 ameliorates nonalcoholic steatohepatitis in mice and macaques. *Science translational medicine* 13: eabg8116

Dear Nobutaka,

Thank you for submitting a revised version of your manuscript. We have now received input from two of the original reviewers, who find that their previous concerns have been addressed satisfactorily and recommend acceptance of the manuscript. There now remain only a few editorial points that need addressing before I can extend official acceptance of the manuscript:

1. Please submit up to five keywords.
2. Please make sure that the order of the sections in the manuscript is as follows: abstract, introduction, results, discussion, materials & methods, data availability section, acknowledgments, disclosure statement and competing interests, references, main figure legends, tables, expanded figure legends.
3. Please upload the main and EV figures as individual production quality figure files in the .eps, .tif, or .jpg format (one file per figure).
4. CRediT has replaced the traditional author contributions section because it offers a systematic, machine-readable author contributions format that allows for more effective research assessment. Please remove the Authors Contributions from the manuscript and use the free text boxes beneath each contributing author's name in our online submission system to add specific details on the author's contribution. More information is available in our guide to authors.
5. Please rename "Competing Interest Statement" section into "Disclosure and competing interests statement" (further info: <https://www.embopress.org/page/journal/14602075/authorguide#conflictsofinterest>).
6. Please remove the EV tables from the manuscript text and upload as two separate files.
7. There is a reference to "data not shown" on page 8. Since our policy does not permit references to "data not shown", please add this data in the Appendix.
8. Please remove the list of abbreviations from the manuscript text file.
9. We noticed the following missing or non-matching items in the source data that you kindly provided:
 - The following source data are not available: Fig. 1D; 2D (Mut/Mut High Mag Masson staining).
 - In Figures 4B (GST/Coomassie staining), 6B (ACC1 panel), 6F (IP: ACC1, Ubq panel), 6H (ACC1), the source data does not appear to match the figure panel. Please check, it is possible that I am not able to locate the right bands due to contrast differences.
 - For figure 3D, the comparison of the source data and the figure panel for GAPDH revealed that the figure panel appears rather strongly contrasted. Please replace the panel with a version that more closely matches the signal range presented in the Western blot.
10. In our standard source data check, we have noted unexplained numerical duplications in the source data for figure 3C. I have attached the corresponding file with the detected duplications labelled in colour. Please take a look and correct if needed. A brief explanation would be very helpful.
11. Our data editors have flagged the following issues in figure legends that need correcting:
 - Please provide the exact p values in the legends of figures 3c, j; 4d-e; 5c-d, g; 6c-d; EV 2e; EV 3b, f; EV 5f-h; EV 7b-c.
 - Please note that in figures 5c-d, f; there is a mismatch between the annotated p values in the figure legend and the annotated p values in the figure file that should be corrected.
 - In the legend for figure 8c, please include the p value for "****" in the figure or legend.
 - Please provide information on the nature and number of replicates in the legends of figures EV 7a-n.
 - Please describe the nature of replicates (e.g., biological or technical) in the legends of figures 2e-h; 3c, j; 4d-e; 5f; 6a, c-e, g, i; 8c-d; EV 3b; EV 5f-h; EV 6b.
 - Please define the error bars in the legend of figure 5f.

Finally, we would like to propose some edits in the title, abstract and synopsis, mainly in order to increase the accessibility of your findings to our more general readership. We have also prepared a short blurb that will accompany the title of your manuscript in our online table of contents. Please take a look at the proposed textual changes in the attached text file and let me know if any corrections are needed.

With best wishes,

Ieva

Ieva Gailite, PhD
Senior Scientific Editor
The EMBO Journal

Meyerhofstrasse 1
D-69117 Heidelberg
Tel: +4962218891309
i.gailite@embojournal.org

Further information is available in our Guide For Authors: <https://www.embopress.org/page/journal/14602075/authorguide> We realize that it is difficult to revise to a specific deadline. In the interest of protecting the conceptual advance provided by the work, we recommend a revision within 3 months (4th Mar 2025). Please discuss the revision progress ahead of this time with the editor if you require more time to complete the revisions.

Referee #1:

I think the authors have made reasonable efforts to answer my comments. Although there are still some minor issues, I would not want those to stop the publication of this paper.

Referee #3:

My original comments were not extensive and the authors have addressed these adequately. The story is solid.

The authors addressed the remaining editorial issues.

Dear Nobutaka,

I sincerely apologise for the delay in communicating the decision due to the holiday period. Thank you for addressing most of the final editorial points. I am now pleased to inform you that your manuscript has been accepted for publication.

Before we forward your manuscript to our publishers, I would still need to include the information on the error bars used in figure 5f. Is it SEM as indicated for the other panels? I can add this information in the manuscript on your behalf.

If you have any questions, please do not hesitate to contact the Editorial Office. Thank you for this contribution to The EMBO Journal and congratulations on a nice study!

With best wishes,

Ieva
